

# A compilation of global bio-optical in situ data for ocean-colour satellite applications – version two

André Valente[1], Shubha Sathyendranath[2], Vanda Brotas[1,2], Steve Groom[2], Michael Grant[2,3], Malcolm Taberner[3], David Antoine[4,5], Robert Arnone[6], William M. Balch[7], Kathryn Barker[8,9,10], Ray Barlow[11], Simon Bélanger[12], Jean-François Berthon[13], Şükrü Beşiktepe[14], Yngve Borsheim[15], Astrid Bracher[16,17], Vittorio Brando[9,18], Elisabetta Canuti[13], Francisco Chavez[19], Andrés Cianca[20], Hervé Claustre[4], Lesley Clementson[9], Richard Crout[21], Robert Frouin[22], Carlos García-Soto[23,24], Stuart W. Gibb[25], Richard Gould[21], Stanford B. Hooker[26], Mati Kahru[22], Milton Kampel[27], Holger Klein[28], Susanne Kratzer[29], Raphael Kudela[30], Jesus Ledesma[31], Hubert Loisel[32], Patricia Matrai[7], David McKee[33], Brian G. Mitchell[22], Tiffany Moisan[34], Frank Muller-Karger[35], Leonie O'Dowd[36], Michael Ondrusek[37], Trevor Platt[2], Alex J. Poulton[38], Michel Repecaud[39], Thomas Schroeder[9], Timothy Smyth[2], Denise Smythe-Wright[40], Heidi M. Sosik[41], Michael Twardowski[42], Vincenzo Vellucci[4], Kenneth Voss[43], Jeremy Werdell[26], Marcel Wernand[44], Simon Wright[45], Giuseppe Zibordi[13]

[1] {MARE - Marine and Environmental Sciences Centre, Faculdade de Ciências, Universidade de Lisboa, Campo Grande, 1749-016 Lisboa, Portugal}

[2] {Plymouth Marine Laboratory, Plymouth, PL1 3DH, UK}

[3] {EUMETSAT, Eumetsat-Allee 1, 64295 Darmstadt, Germany}

[4] {Sorbonne Université, CNRS, Laboratoire d'Océanographie de Villefranche, LOV, F-06230 Villefranche-sur-Mer, France}

[5] {Remote Sensing and Satellite Research Group, School of Earth and Planetary Sciences, Curtin University, Perth, WA 6845, Australia}

[6] {University of Southern Mississippi, Stennis Space Center, MS, USA}

[7] {Bigelow Laboratory for Ocean Sciences, 60 Bigelow Dr., East Boothbay, ME 04544, Maine, USA}

[8] {ARGANS Ltd, UK}

[9] {CSIRO Oceans and Atmosphere, Australia}

[10] {Australian Research Data Commons}

[11] {Bayworld Centre for Research and Education, Cape Town, South Africa}

[12] {Université du Québec à Rimouski, Rimouski (Québec), Canada}

[13] {European Commission, Joint Research Centre, Ispra, Italy}

[14] {Dokuz Eylul University, Institute of Marine Science and Technology, Izmir, Turkey}

[15] {Institute of Marine Research, Bergen, Norway}

[16] {Alfred-Wegener-Institute Helmholtz Centre for Polar and Marine Research, Bremerhaven, Germany}

[17] {Institute of Environmental Physics, University Bremen, Bremen, Germany}



[18] {CNR - ISMAR, Rome, Italy}

[19] {Monterey Bay Aquarium Research Institute, Moss Landing, CA, USA}

[20] {PLOCAN-Oceanic Platform of the Canary Islands. Carretera de Taliarte, 35214 Telde, Gran Canaria, Spain}

[21] {Naval Research Laboratory, Stennis Space Center, MS, USA}

[22] {Scripps Institution of Oceanography, University of California San Diego, CA, USA}

[23] {Spanish Institute of Oceanography (IEO), Corazón de María 8, 28002 Madrid, Spain}

[24] {Plentziako Itsas Estazioa/ Euskal Herriko Unibetsitatea (PIE/EHU), Areatza z/g, 48620 Plentzia, Spain}

[25] {Environmental Research Institute, North Highland College, University of the Highlands and Islands, Thurso, Scotland, UK}

[26] {NASA Goddard Space Flight Center, Greenbelt, Maryland, USA}

[27] {Remote Sensing Division, National Space Research Institute (INPE), Sao Jose dos Campos, Brazil}

[28] {Operational Oceanography Group, Federal Maritime and Hydrographic Agency, Hamburg, Germany}

[29] {Department of Ecology, Environment and Plant Sciences, Stockholm University, 106 91 Stockholm, Sweden}

[30] {University of California Santa Cruz, Santa Cruz, CA USA}

[31] {Instituto del Mar del Perú}

[32] { Laboratoire d'Océanologie et de Géosciences, Université du Littoral-Côte-d'Opale, Université Lille, CNRS, UMR 8187, LOG, 32 avenue Foch, Wimereux, France }

[33] {Physics Dept, University of Strathclyde, Glasgow, G4 0NG, Scotland}

[34] {NASA Goddard Space Flight Center, Wallops Flight Facility, Wallops Island, VA, USA}

[35] {Institute for Marine Remote Sensing/ImaRS, College of Marine Science, University of South Florida, FL, USA}

[36] {Fisheries and Ecosystem Advisory Services, Marine Institute, Rinville – Oranmore, Galway, Ireland}

[37] {NOAA/NESDIS/STAR/SOCD, College Park, MD, USA}

[38] {Lyell Centre for Earth and Marine Science and Technology, Heriot-Watt University, Edinburgh, UK}

[39] {IFREMER Centre de Brest, Plouzane, France}

[40] {Ocean Biogeochemistry and Ecosystems, National Oceanography Centre, Waterfront Campus, Southampton, UK}

[41] {Biology Department, Woods Hole Oceanographic Institution, Woods Hole, MA, USA}

[42] {Harbor Branch Oceanographic Institute, Fort Pierce, FL, USA}

[43] {University of Miami, Coral Gables, FL, USA}

[44] {Royal Netherlands Institute for Sea Research, Texel, Netherlands}

[45] {Australian Antarctic Division and the Antarctic Climate and Ecosystems Cooperative Research Centre, Hobart, Australia}

*Correspondence to*: A. Valente (adovalente@fc.ul.pt)



**Abstract.** A global compilation of in situ data is useful to evaluate the quality of ocean-colour satellite data records. Here we describe the data compiled for the validation of the ocean-colour products from the ESA Ocean Colour Climate Change Initiative (OC-CCI). The data were acquired from several sources (including, *inter alia*, MOBY, BOUSSOLE, AERONET-OC, SeaBASS, NOMAD, MERMAID, AMT, ICES, HOT, GeP&CO) and span the period from 1997 to 2018. Observations of the following variables were compiled: spectral remote-sensing reflectances, concentrations of chlorophyll-a, spectral inherent optical properties, spectral diffuse attenuation coefficients and total suspended matter. The data were from multi-project archives acquired via open internet services or from individual projects, acquired directly from data providers. Methodologies were implemented for homogenisation, quality control and merging of all data. No changes were made to the original data, other than averaging of observations that were close in time and space, elimination of some points after quality control and conversion to a standard format. The final result is a merged table designed for validation of satellite-derived ocean-colour products and available in text format. Metadata of each in situ measurement (original source, cruise or experiment, principal investigator) were propagated throughout the work and made available in the final table. By making the metadata available, provenance is better documented, and it is also possible to analyse each set of data separately. This paper also describes the changes that were made to the compilation in relation to the previous version (Valente et al., 2016). The compiled data are available at https://doi.org/10.1594/PANGAEA.898188 .

# 1 Introduction

Currently, there are several sets of in situ bio-optical data, worldwide, suitable for validation of ocean-colour satellite data. Although some are managed by the data producers, others are in international repositories with contributions from multiple scientists. Many have rigid quality controls and are built specifically for ocean colour validation. The use of only any one of these data sets would limit the number of data in validation exercises. It is, therefore, vital to acquire and merge all these data sets into a single unified data set to maximize the number of matchups available for validation, their distribution in time and space, and, consequently, to reduce uncertainties in the validation exercise. However, merging several data sets together can be a complicated task. First it is necessary to acquire and harmonize all data sets into a single standard format. Second, during the merging, duplicates between data sets have to be identified and removed. Third, the metadata should be propagated throughout the process and made available in the final merged product. Ideally, the compiled data set would be made available as a simple text table, to facilitate ease of access and manipulation. In this work such unification of multiple data sets is presented. This was done for the validation of the ocean-colour products from the ESA Ocean Colour Climate Change Initiative (OC-CCI), but with the intent to serve the broader user community as well.

A merged data set is not without drawbacks: it is likely to be large and so not always easy to manipulate; because the merging is done on pre-existing, processed databases, it is not possible to have full control of the whole processing chain; the data set would be a compilation of observations collected by several investigators using different instruments, sampling methods and protocols, which might eventually have been modified by the processing routines used by the repositories or





archives. To minimise these potential drawbacks, we have, for the most part, incorporated only data sets that have emerged from the long-term efforts of the ocean-colour and biological oceanographical communities to provide scientists with high-quality in situ data, and implemented additional quality checks on the data, to enhance confidence in the quality of the merged product. Nevertheless, it is still recognized that different and somehow unpredictable uncertainties may affect data from the diverse sources as a result of the application of a variety of field/laboratory instruments, methods and data reductions schemes.

In Sect. 2 the methodologies used to harmonize and integrate all data, as well as a description of individual data sets acquired are provided. In Sect. 3 the geographic distribution and other characteristics of the final merged data set are shown. Section 4 provides an overview of the data.

## 2 Data and methods

### 2.1 Pre-processing and merging

The compiled global set of bio-optical in situ data described in this work has an emphasis, though not exclusively, on open-ocean data. It comprises the following variables: remote-sensing reflectance ("rrs"), chlorophyll-a concentration ("chla"), algal pigment absorption coefficient ("aph"), detrital and coloured dissolved organic matter absorption coefficient ("adg"), particle backscattering coefficient ("bbp"), diffuse attenuation coefficient for downward irradiance ("kd") and total suspended matter ("tsm"). The variables "rrs", "aph", "adg", "bbp" and "kd" are spectrally dependent, and this dependence is, hereafter, implied. The data were compiled from 27 sources (MOBY, BOUSSOLE, AERONET-OC, SeaBASS, NOMAD, MERMAID, AMT, ICES, HOT, GeP&CO, AWI, ARCSSPP,BARENTSSEA, BATS, BIOCHEM, BODC, CALCOFI, CCELTER, CIMT, COASTCOLOUR, ESTOC, IMOS, MAREDAT, PALMER, SEADATANET, TPSS and TARA): each one described in Sect. 2.2. The data sources in this work should also be viewed as groups of data that were acquired from a specific source, standardized with a specific method and later merged into the compilation. The compiled in situ observations have a global distribution and cover the period 1997 to 2018. The listed variables, with the exception of total suspended matter, were chosen as they are the operational satellite ocean-colour products of ESA OC-CCI project, which currently focuses on the merging of four ocean-colour satellite sensors: the Medium Resolution Imaging Spectrometer (MERIS) of ESA; the Moderate Resolution Imaging Spectro-radiometer (MODIS) of NASA; the Sea-viewing Wide Field-of-view Sensor (SeaWiFS) of NASA; and the Visible Infrared Imaging Radiometer Suite (VIIRS) of NASA and National Oceanic and Atmospheric Administration (NOAA), to create a time series of satellite data.

This is a second version of the compilation of global bio-optical in situ data described by Valente et al. (2016). The new version has more data, and a higher temporal and spatial coverage. The increases in number of observations are mainly for "chla", "rrs" and "aph". In comparison with Valente et al. (2016), the observations of "chla" and "aph" have doubled in number and provide a better spatial coverage, especially in the Southern and Arctic Oceans. The "rrs" values also increased



in number, but not as much in spatial coverage, because most of the new observations came from fixed locations.

The present second version is a compilation of data from sources used in the first version (MOBY, BOUSSOLE, AERONET-OC, SeaBASS, NOMAD, MERMAID, AMT, ICES, HOT, GeP&CO) plus data from the new sources (AWI, ARCSSPP, BARENTSSEA, BATS, BIOCHEM, BODC, CALCOFI, CCELTER, CIMT, COASTCOLOUR, ESTOC,

IMOS, MAREDAT, PALMER, SEADATANET, TPSS and TARA). The main differences to the first version are: 1) some of the data sources used in the first version were updated (MOBY, AERONET, SeaBASS and HOT), 2) new data sources were added, 3) a new variable was compiled (Total Suspended Matter), 4) the format in which the compilation is provided is different, and 5) two flags were added.

Concerning the change of format, in Valente et al. ( 2016) the compilation was provided as one unique two-dimensional

table. Now, given its increased size (136,250 rows and 1,286 columns compared with 80,524 rows and 267 columns previously), the table has been broken into three smaller tables that relate to each other via one unique key identifying each row. One additional table is also provided to help with data manipulation. Despite this change, the compilation should still be viewed conceptually as one unique table, and as such, it is still described in that way. In the present version, two flags were added: "flag_time" and "flag_chl_method". The first is because in the present version three data sources were used (ESTOC,

MAREDAT and TPSS) where information on time (hour of the day) was not available. The time for these observations was set to 12:00:00 (UTC) and the observations were flagged with "1" in column "flag_time".  A second flag was necessary, because in two data sources (ARCSSPP and SEADATANET) there was uncertainty on whether the compiled chlorophyll were from fluorometry, spectrophotometric or HPLC methods. The compiled chlorophyll observations from these two data sources were flagged with "1" in column "flag_chl_method" and were marked as "chla_fluor".

Remote-sensing reflectance ("rrs") is a primary ocean colour product defined as "rrs = Lw/Es", where "Lw" is the upward water-leaving radiance and "Es" is the total downward irradiance at sea level. Remote-sensing reflectance is related to irradiance reflectance ("Rw") approximately through "rrs = Rw/Q", where Q ranges from 3 to 5 in natural waters and is equal to $\pi$ for an isotropic (Lambertian) light field. Another quantity that is often required is the "normalized" water-leaving radiance ("nLw") (Gordon and Clark, 1981), which is related to remote-sensing reflectance via "rrs = nLw/Fo", where "Fo"

is the top-of-the-atmosphere solar irradiance. If not directly available, remote-sensing reflectance was calculated through the equations described above, depending on the format of the original data. The original data were acquired in an advanced form (e.g. time-averaged, extrapolated to surface), from nine data sources designed for ocean-colour validation and applications (MOBY, BOUSSOLE, AERONET-OC, SeaBASS, NOMAD, MERMAID, COASTCOLOUR, TARA, AWI), therefore, only requiring the conversion to a common format. In the processing made by the space agencies, the quantity

"rrs" is normalized to a single Sun-viewing geometry (Sun at zenith and nadir viewing) taking in account the bidirectional effects as described in Morel and Gentili (1996) and Morel et al. (2002). Thus, for consistency with satellite "rrs" product, only in situ "rrs" that included the latter normalization were included in the compilation.

Chlorophyll-a concentration is the conventional measure for phytoplankton biomass and one of the most-widely used





satellite ocean-colour products (IOCCG, 2008). To validate satellite-derived chlorophyll-a concentration, two different variables were compiled: one of these represents chlorophyll-a measurements made through fluorometric or spectrophotometric methods, referred to hereafter as "chla_fluor" and the other is the chlorophyll concentration derived from HPLC (High-Performance Liquid Chromatography) measurements, referred to hereafter as "chla_hplc". The chlorophyll

data were compiled from the following 25 data sources: BOUSSOLE, SeaBASS, NOMAD, MERMAID, AMT, ICES, HOT, GeP&CO, AWI, ARCSSPP, BARENTSSEA, BATS, BIOCHEM, BODC, CALCOFI, CCELTER, CIMT, COASTCOLOUR, ESTOC, IMOS, MAREDAT, PALMER, SEADATANET, TPSS and TARA. One requirement for "chla_fluor" measurements was that they were made using in vitro methods (i.e. based on extractions of chlorophyll-a). Although this severely decreased the number of observations, since in situ fluorometry (e.g. fluorometers mounted on

CTD's) is widely available in oceanographic databases, it was decided to exclude such data because of potential problems with the calibration of in situ fluorometers. The variable "chla_hplc" was calculated by summing all reported chlorophyll-a derivatives, including divinyl chlorophyll-a, epimers, allomers, and chlorophyllide-a. The two chlorophyll variables are retained separately in the database to facilitate their use. HPLC measurements could be considered of higher quality, but fluorometric measurements are more numerous. Thus one option for users is to use "chla_fluor" only when there are no

"chla_hplc" measurements available. To be consistent with satellite-derived chlorophyll values, which are derived from the light emerging from the upper layer of the ocean, all chlorophyll observations in the top 10 meters (replicates at the same depth, or measurements at multiple depths) were averaged if the coefficient of variation among observations was less than 50 %, otherwise they were discarded. The averages were then assigned to the surface. The depth of 10 m was chosen as a compromise between clear oligotrophic and turbid eutrophic waters. Other methods, such as chlorophyll depth-averages

using local attenuation conditions (Morel and Maritorena, 2001), require observations at multiple depths, which, given our decision to use only in vitro measurements, would have reduced considerably the final number of observations.

With regard to the inherent optical properties ("aph", "adg", "bbp"), if not already calculated and provided in the contributed data sets, they were computed from related variables that were available: particle absorption ("ap"), detrital absorption ("ad"), coloured dissolved organic matter (CDOM) absorption ("ag"), total backscattering ("bb"). The following equations

were used "adg = ad + ag", "ap = aph + ad", and "bb = bbp + bbw". For the latter equation, the variable "bbw" was computed using "bbw = bw/2", where "bw" is the scattering coefficient of seawater derived from Zhang et al. (2009). The diffuse attenuation coefficient for downward irradiance ("kd") did not require any conversion and was compiled as originally acquired. Observations of inherent optical properties (surface values) and diffuse attenuation coefficient for downward irradiance, were acquired in total from six data sources designed for ocean-colour validation and applications (SeaBASS,

NOMAD, MERMAID, AWI, COASTCOLOUR, TPSS), thus already subject to the processing routines of these data sets. Concerning total suspended matter, these data were compiled as originally available from MERMAID and COASTCOLOUR.

The merged data set was compiled from 27 sets of in situ data, which were obtained individually either from archives that



incorporate data from multiple contributors (SeaBASS, NOMAD, MERMAID, ICES, ARCSSPP, BIOCHEM, BODC, COASTCOLOUR, MAREDAT, SEADATANET), or from particular contributors, measurement programs or projects (MOBY, BOUSSOLE, AERONET-OC, HOT, GeP&CO, AMT, AWI, BARENTSSEA, BATS, CALCOFI, CCELTER, CIMT, ESTOC, IMOS, PALMER, TPSS, TARA) and were subsequently, homogenized and merged. Data contributors are listed in Table 2 and in the auxiliary material. There were methodological differences between data sets. Therefore, after acquisition, and prior to any merging, each set of data was pre-processed for quality control and converted to a common format. During this process, data were discarded if they had: 1) unrealistic or missing, date and geographic coordinate fields; 2) poor quality (e.g. original flags) or method of observation that did not meet the criteria for the data set (e.g. in situ fluorescence for chlorophyll concentration); and 3) spuriously high or low data. For the last, the following limits were imposed: for "chla_fluor" and "chla_hplc" [0.001-100] mg m$^{-3}$; for "rrs" [0-0.15] sr$^{-1}$; for "aph", "adg" and "bbp" [0.0001-10] m$^{-1}$; for "tsm" [0-1000] g m$^{-3}$; for "kd" [(aw($\lambda$)-10] m$^{-1}$, where "aw" are the pure water absorption coefficients derived from Pope and Fry (1997). Also during this stage, three metadata strings were attributed to each observation: "dataset", "subdataset" and "contributor". The "dataset" contains the name of the original set of data, and can only be one of the following: "aoc", "boussole", "mermaid", "moby", "nomad", "seabass", "hot", "ices", "amt", "gepco", "arcsspp", "awi", "barentssea", "bats", "biochem", "bodc", "calcofi", "cc", "ccelter", "cimt", "estoc", "imos", "maredat", "palmer", "seadatanet", "tpss", "tara". The "subdataset" starts with the "dataset" identifier and is followed by additional information about the data, as <dataset>_<cruise/station/site>) (e.g. "seabass_car81"). The "contributor" contains the name of the data contributor. An effort was made to homogenize the names of data contributors from the different sets of data. These three metadata are the link to trace each observation to its origin and were propagated throughout the processing. Finally, this processing stage ended with each set of data being scanned for replicate variable data and replicate station data, which when found, were averaged if the coefficient of variation was less than 50 %, otherwise they were discarded. Replicates were defined as multiple observations of the same variable, with the same date, time, latitude, longitude and depth. Replicate station data were defined as multiple measurements of the same variable, with the same date, time, latitude and longitude. For the latter case, a search window of 5 minutes in time and 200 meters in distance was given, to account for station drift. A small number of observations that were identified as replicates had a different "subdataset" identifiers (i.e. different cruise names). These observations were considered suspicious if the values were different, and discarded. If the values were the same, one of the observations was retained. This possibly originated from the same group of data being contributed to an archive by two different data contributors.

Once a set of data was homogenized, its data were integrated into a unique table. This final merging focused on the removal of duplicates between the sets of data. Although some duplicates are known (e.g. MOBY, BOUSSOLE, AERONET-OC and NOMAD data are found in SeaBASS and MERMAID), others are unknown (e.g. how many of GeP&CO, ICES, AMT, HOT are within NOMAD, SeaBASS and MERMAID). Therefore, duplicates were identified using the metadata ("dataset" and "subdataset") when possible, and temporal-spatial matches, as an additional precaution. For temporal-spatial matches, several thresholds were used, but typically 5 minutes and 200 meters were taken to be sufficient to identify most duplicated



data, which reflected small differences in time, latitude and longitude, between the different sets of data. Larger thresholds were used in some cases as a cautionary procedure. This was the case when searching for NOMAD data in other data sets, because NOMAD includes a few cases where merging of radiometric and pigment data was done with large spatial-temporal thresholds (Werdell and Bailey, 2005). A large temporal threshold was also used when integrating observations from the

three data sources that did not have time available (ESTOC, MAREDAT and TPSS). In regard to all data, if duplicates were found, data from the NOMAD data set were selected first, followed by data from individual projects or contributors (MOBY, BOUSSOLE, AERONET-OC, AMT, HOT,GeP&CO, AWI, BARENTSSEA, BATS, CALCOFI, CCELTER, CIMT, ESTOC, IMOS, PALMER, TPSS and TARA) and finally for the remaining data sets (SeaBASS, MERMAID,ICES, ARCSSPP, BIOCHEM, BODC, COASTCOLOUR, MAREDAT and SEADATANET). This procedure was chosen to

preserve the NOMAD data set as a whole, since it is widely used in ocean-colour validation. It should be noted that, by this procedure, data from individual projects or contributors may be listed under NOMAD (e.g. some PALMER data are found in NOMAD with metadata string "nomad_palmer_lter"). After giving priority to NOMAD, the priority was generally given to data from individual projects or contributors, but due to an incremental approach, where only new data are added to previous versions of the compilation, some data from individual projects or contributors (BATS, CALCOFI, CIMT, PALMER and

TPSS) added in later stages, may be found under other data sources. This occurs mainly for BATS and CALCOFI, which have their earlier chlorophyll data in SeaBASS with metadata strings "seabass_bats*" and "seabass_cal*", and also CIMT which has some of its data under COASTCOLOUR. After all data from a given source were free of duplicates, they were merged consecutively by variable in the final table. During this process, we also searched for rows (stations) that were separated from each other by time differences less than 5 minutes and horizontal spatial differences of less than 200 meters.

When such rows were found, the observations in those rows were merged into a single row. The compiled merged data were compared with the original sets to certify that no errors occurred during the merging. As a final step, a water-column (station) depth was recorded for each observation, which was the closest water column depth from the ETOPO1 global relief model (National Geophysical Data Center ETOPO1; Amante and Eakins, 2009). For observations where the closest water depth was above sea level (e.g. data collected very near the coast), it was given the value of zero.

Data processing thus included two major steps: pre-processing and merging. The first step was related to each set of contributing data sets in particular and aimed to identify problems and convert the data of interest to a standard format. The second step dealt with the integration of data into one unique file and included the elimination of duplicated data between the individual sets of data. In the next subsections a brief overview of each original set of data is provided.

## 2.2 Pre-processing of each set of data

### 2.2.1 Marine Optical Buoy (MOBY)

MOBY is a fixed mooring system operated by the National Oceanic and Atmospheric Administration (NOAA) that provides a continuous time series of water-leaving radiance and surface irradiance in the visible region of the spectra since 1997. The



site is located a few kilometres west of the Hawaiian Island of Lanai where the water depth is about 1200 m. Since its deployment, MOBY measurements have been the primary basis for the on-orbit vicarious calibrations of the SeaWiFS and MODIS ocean colour sensors. A full description of the MOBY system and processing is provided in Clark et al. (2003). Data are freely available for scientific use at the MOBY Gold directory. The products of interest are the "Scientific Time Series"

files, which refer to MOBY data averaged over sensor-specific wavelengths and particular hours of the day (around 20-23 UTC). For this work, the satellite band-average products for SeaWiFS, MODIS AQUA, MERIS, VIIRS and the Ocean and Land Colour Instrument (OLCI) were compiled from the "R2017 Reprocessing" . The "inband" average subproduct was used, and to maintain the highest quality, only data determined from the upper two arms ("Lw1") and flagged "good" quality were acquired. Data from the MOBY203 deployment were discarded due to the absence of surface irradiance data. The

compiled variable was the remote-sensing reflectance, "rrs", which was computed from the original water-leaving radiance ("Lw") and surface irradiance ("Es"). The water-leaving radiances were corrected for the bidirectional nature of the light field (Morel and Gentili, 1996; Morel et al., 2002) using the same look-up table and method as that used in the SeaWiFS Data Analysis System (SeaDAS) processing code. The MOBY data were reprocessed in 2017 ("MOBY R2017 Reprocessing") to include various improvements in the calibration of the instrument and post processing, which include: 1) a

new method to extrapolate the upwelling radiance attenuation coefficient to the surface; 2) an increase in arm depth by 0.234 m; and 3) a single pixel shift in the data for the red spectrograph collected at a bin factor of 384. Only the last two changes were included in present compilation. As mentioned before, the MOBY data compiled in this work are sensor-specific. Therefore, attention is necessary to use the correct MOBY data when validating a particular sensor. The way MOBY data are stored in the final merged table is consistent with the original wavelengths; however, these wavelengths can differ from what

is sometimes expected to be the central wavelength of a given band and sensor. Irrespective of the wavelength where MOBY data are stored in the final table, for validation of bands 1-6 of SeaWiFS, MOBY data stored in the final merged table at 412, 443, 490, 510, 555 and 670 nm, respectively, should be used. For validation of bands 1-6 of MODIS AQUA, MOBY data stored in the final merged table at 416, 442, 489, 530, 547 and 665 nm, respectively, should be used. For validation of bands 1-7 of MERIS, MOBY data stored in the final merged table at 410.5, 440.4, 487.8, 507.7, 557.6, 617.5 and 662.4 nm,

respectively, are the appropriate data. For validation of bands 2-8 of OLCI, MOBY data stored in the final merged table at 412.0676, 443.1898, 490.7176, 510.6403, 560.5796, 620.626 nm and 665.3737, respectively, are the appropriate data. Finally, for validation of bands 1-5 of VIIRS, MOBY data stored in the final merged table at 412.9, 444.5, 481.2, 556.3 and 674.6 nm, respectively, are the appropriate data.

**2.2.2 BOUée pour l'acquiSition de Séries Optiques à Long termE (BOUSSOLE)**

BOUSSOLE Project started in 2001 with the objective of establishing a time series of bio-optical properties in oceanic waters to support the calibration and validation of ocean-colour satellite sensors (Antoine et al., 2006). The project consists of a monthly cruise program and a permanent optical mooring (Antoine et al., 2008). The mooring collects radiometry and inherent optical properties (IOPs) in continuous mode every 15 minutes at 2 depths (4 and 9 m nominally). The monthly





cruises are devoted to the mooring servicing, to the collection of vertical profiles of radiometry and IOPs, and to water sampling at 11 depths from the surface down to 200 m, for subsequent analyses including phytoplankton pigments, particulate absorption, CDOM absorption and suspended particulate matter load. The BOUSSOLE mooring is in the Western Mediterranean Sea at a water depth of 2400m. All pigment (2001-2012) and radiometric (2003-2012) data were provided by
the Principal Investigator. The compiled variables were "rrs" and "chla_hplc". Observations of the diffuse attenuation coefficient ("kd") were not included in the present compilation, as they were under internal quality revision at the time of data acquisition. Remote-sensing reflectance was computed from the original "fully-normalized" water-leaving radiance ("nLw_ex"), which is the "normalized" water-leaving radiance ("nLw" previously described), with a correction for the bidirectional nature of the light field (Morel and Gentili, 1996; Morel et al., 2002). The solar irradiance ("Fo") was computed
from two available variables in the original set of data: the normalized water-leaving radiance ("nLw") and the remote-sensing reflectance ("rrs"), using the equation "Fo = nLw/rrs". Only radiometric observations that meet the following criteria were used: 1) tilt of the buoy was less than 10 °; 2) the buoy was not lowered by more than 2 m as compared to its nominal water line (to ensure the Es reference sensor is above water and exempt from sea spray); and 3) the solar irradiance was within 10 % of its theoretical clear-sky value (determined from Gregg and Carder, 1990). The latter criterion was used to
select clear skies only. An additional quality control was to remove observations that were 50 % higher or lower than the daily average. This removed a small number of "spikes" in the time series. The final quality control step was to remove days where the standard deviation was more than half of the daily average. This was meant to identify days with high variability. Very few days (N = 2) were removed with this test. These quality control criteria were applied per wavelength, which resulted in some observations with an incomplete spectrum.

**2.2.3 AErosol RObotic NETwork-Ocean Color (AERONET-OC)**

AERONET-OC is a component of AERONET, including sites where sun-photometers operate with a modified measurement protocol leading to the determination of the fully-normalized water leaving radiance (Zibordi et al., 2006; Zibordi et al., 2009). As a result of collaboration between the Joint Research Centre (JRC) and NASA, this component has been specifically developed for the validation of ocean-colour radiometric products. The strength of AERONET-OC is "the
production of standardised measurements that are performed at different sites with identical measuring systems and protocols, calibrated using a single reference source and method, and processed with the same codes" (Zibordi et al., 2006; Zibordi et al., 2009). All high quality data ("Level-2") were acquired from the project website, for 11 sites: Abu_Al_Bukhoosh (~25° N, ~53° E) , COVE_SEAPRISM (~36° N, ~75° W), Gloria (~44° N, ~29 °E), Gustav_Dalen_Tower (~58 °N, ~17 °E), Helsinki Lighthouse (~59 °N, ~24° E), LISCO (~40° N, ~73° W), Lucinda (~18° S,
~146° E), MVCO (~41° N, ~70° W), Palgrunden (~58° N, ~13° E; Philipson et al., 2016), Venice (~45° N, ~12° E) and WaveCIS_Site_CSI_6 (~28° N, ~90° W). The compiled variable was "rrs". Remote-sensing reflectance was computed from the original "fully-normalized" water-leaving radiance (see Sect. 2.2.2 for definition). The solar irradiance ("Fo"), which is not part of the AERONET-OC data, was computed from the Thuillier (2003) solar spectrum irradiance, by averaging "Fo"



over a wavelength-centred 10 nm window. Data were compiled for the exact wavelengths of each record, which can change over time for a given site depending on the specific instrument deployed.

In comparison with the previous compilation of AERONET-OC data from Lucinda site a calibration correction was applied by NASA affecting instrument SN-520. All radiometric data from this instrument provided by NASA prior to October 2018

were underestimated by approximately a factor of two due to incorrect application of instrument gains during the processing.

### 2.2.4 SeaWiFS Bio-optical Archive and Storage System (SeaBASS)

SeaBASS is one of the largest archives of in situ marine bio-optical data (Werdell and Bailey, 2003). It is maintained by NASA's Ocean Biology Processing Group (OBPG) and includes measurements of optical properties, phytoplankton pigment concentrations, and other related oceanographic and atmospheric data. The SeaBASS database consists of in situ data from

multiple contributors, collected using a variety of measurement instruments with consistent, community-vetted protocols, from several marine platforms such as fixed buoys, hand-held radiometers and profiling instruments. Quality control of the received data includes a rigorous series of protocols that range from file format verification to inspection of the geophysical data values (Werdell and Bailey, 2003). Radiometric data were acquired through the "Validation" search tool, which provided in situ data with matchups for particular ocean-colour sensors (Bailey and Werdell, 2006). The criterion in the

search-query was defined to have the minimal flag conditions in the satellite data, to retrieve a greater number of matchups and, therefore, in situ data. Regarding phytoplankton pigment data, the majority were acquired through the "Pigment" search tool, which provided pigment data directly from the archives. As was stated in the SeaBASS website , the "Pigment" search tool was originally designed to return only in vitro fluorometric measurements, which is consistent with our approach, but over time chlorophyll-a measurements made using other methods (e.g. in situ fluorometry) were included in the retrieved

pigment data. In the pigment data used in this work, a large number of in situ fluorometric measurements from continuous underway instruments were identified and discarded. These data were initially identified from cruises with more than 50 observations per day, and then re-checked in the SeaBASS website to confirm whether indeed they were continuous underway measurements. A total of 120412 such measurements were identified and discarded. Given the large volume of this group of data, it is possible that some chlorophyll-a observations from in situ methods may have escaped the scrutiny

and persited into the final merged data set. The "Pigment" search tool was recently discontinued and, instead, the "File" search tool can be used, which was also used here to acquire chlorophyll observations for more recent years. The compiled variables from SeaBASS data were: "rrs", "chla_hplc", "chla_fluor", "aph", "adg", "bbp", "kd". No conversion was necessary since all variables were acquired in the desired format.

### 2.2.5 NASA bio-Optical Marine Algorithm Data set (NOMAD)

NOMAD is a publicly-available data set compiled by the NASA OBPG at the Goddard Space Flight Center. It is a high-quality global data set of coincident radiometric and phytoplankton pigment observations for use in ocean-colour algorithm





development and satellite-data product-validation activities (Werdell and Bailey, 2005). The source bio-optical data is the SeaBASS archive, therefore, many dependencies exist between these two data sets, which were addressed during the merging. The current version (Version 2.0 ALPHA, 2008) includes data from 1991 to 2007 and an additional set of observations of inherent optical properties. The current version was used in this work, but with an additional set of columns

of remote-sensing reflectance corrected for the bidirectional effects (Morel and Gentili, 1996; Morel et al., 2002). This additional set of columns was provided directly by the NOMAD creators. The compiled variables were "rrs", "chla_hplc", "chla_fluor", "aph", "adg", "bbp", "kd". Conversion was necessary only for "aph", "adg" and "bbp", and followed the procedures described in Sect. 2.1. For the calculation of "bbp" the variable "bb" was used with a smooth fitting to remove noise. A portion of NOMAD data were optically weighted (for methods see Werdell and Bailey, 2005). These data are not

consistent with the protocols chosen in this work, but these observations were retained since NOMAD is a widely-used data set in ocean-colour validation.

### 2.2.6 MERIS Match-up In situ Database (MERMAID)

MERMAID provides in situ bio-optical data matched with concurrent and comparable MERIS Level 2 satellite ocean-colour products (Barker, 2013a; Barker, 2013b). The MERMAID in situ database consists of data from multiple contributors,

measured using a variety of instruments and protocols, from several marine platforms such as fixed buoys, hand-held radiometers and profiling instruments. Comprehensive quality control and protocols are used by MERMAID to integrate all the data into a common and comparable format (Barker, 2013a; Barker, 2013b). Access to MERMAID data is limited to the MERIS Validation Team, the MERIS Quality Working Group and to the in situ data contributors. For this work, access has been granted to the MERMAID database, through a signed Service Level Agreement. The MERMAID data includes sub-sets

of several data sets used in this compilation (MOBY, AERONET-OC, BOUSSOLE, NOMAD). These observations were removed from the MERMAID dataset to avoid duplication (as discussed in Sect. 2.1). The compiled variables were "rrs", "chla_hplc", "chla_fluor", "aph", "adg", "bbp", "kd" and "tsm". Remote-sensing reflectance was calculated by dividing by $\pi$ the original irradiance reflectance provided. Conversion was also necessary for "aph", "adg" and "bbp", and followed the procedures described in Sect. 2.1.

### 25 2.2.7 Hawaii Ocean Time-series (HOT)

HOT programme provides repeated comprehensive observations of the hydrography, chemistry and biology of the water column at a station located 100 km north of Oahu, Hawaii, since October 1988 (Karl and Michaels, 1996). This site is representative of the North Pacific subtropical gyre. Cruises are made approximately once a month to the deep-water Station ALOHA (A Long-Term Oligotrophic Habitat Assessment; 22° 45' N, 158° 00' W). Pigment data ("chla_hplc" and

"chla_fluor") were extracted directly from the project website. Radiometric measurements from the HOT project are also available, but observations of "rrs" and "kd" from the HOT project were acquired in this work as part of the SeaBASS data set.



### 2.2.8 Geochemistry, Phytoplankton, and Color of the Ocean (GeP&CO)

GeP&CO is part of the French PROOF programme and aims to describe and understand the variability of phytoplankton populations, and to assess its consequences on the geochemistry of the oceans (Dandonneau and Niang, 2007). It is based on the quarterly travels of the merchant ship Contship London from France to New Caledonia in the Pacific. A scientific
observer sailed on each trip and operated the sampling for surface water, filtration, various measurements and checking at several times of each day. The experiment started in October 1999 and finished in July 2002. Pigment data were extracted from the project website. Additional pigment data obtained during the OISO-4 cruise in the south Indian Ocean onboard R/V Marion-Dufresne (Jan-Feb 2000) were added. The samples were measured by Yves Dandonneau following the method used in the GeP&CO project. The compiled variable was "chla_hplc" and "chla_fluor".

**2.2.9 Atlantic Meridional Transect (AMT)**

AMT is a multidisciplinary programme, which undertakes biological, chemical and physical oceanographic research during an annual voyage between the UK and destinations in the South Atlantic (Robinson et al., 2006). The programme was established in 1995 and since then has completed 28 research cruises. Pigment data between 1997 (AMT5) and 2005 (AMT17) were provided by the British Oceanographic Data Centre (BODC) following a specific request for discrete
observations of chlorophyll-a concentration since 1997. The AMT data were isolated by searching for the string "AMT" in the "Cruise" columns and the respective Principal Investigators were then searched individually in a separated metadata file. Data not flagged with highest quality or without method of measurement were not used. For any interest in the original data, BODC is the point of contact, which ensures that if there are any updates, the most recent data are supplied. The compiled variables are "chla_hplc" and "chla_fluor".

**2.2.10 International Council for the Exploration of the Sea (ICES)**

ICES is a network of more than 4000 scientists from almost 300 institutes, with 1600 scientists participating in activities annually. The ICES Data Centre manages a number of large data set collections related to the marine environment covering the North East Atlantic, Baltic Sea, Greenland Sea and Norwegian Sea. The majority of data originate from national institutes that are part of the ICES network of member countries. Data were provided (on 2014-04-28) from the ICES
database on the marine environment (Copenhagen, Denmark) following a specific request. The ICES data were made available under the ICES data policy and if there is any conflict between this and the policy adopted by the users, then the ICES policy applies. The compiled variables were "chla_hplc" and "chla_fluor".

**2.2.11 Arctic System Science Primary Production (ARCSSPP)**

ARCSSPP database is a synthesis of observations between 1954 and 2006, from the Arctic Ocean and northern Seas (Matrai
et al., 2013). The observations were acquired from data repositories, publications or provided by individual investigators.





The database includes quality-controlled observations of productivity and chlorophyll a, photosynthetically available radiation and hydrographic parameters. This collection of data was acquired at http://www.nodc.noaa.gov/cgi-bin/OAS/prd/accession/download/63065. For the present work, only observations of chlorophyll-a concentration with known time zones were used. The compiled chlorophyll observations were from discrete samples, but the exact method (either "chla_fluor" or "chla_hplc") was not available for all observations. Thus, the ARCSSPP chlorophyll observations were marked as "chla_fluor", although some might have been from HPLC measurements, and were flagged with "1" in a column "flag_chla_method". The compiled variable was "chla_fluor".

### 2.2.12 Data provided by Astrid Bracher, Alfred-Wegener-Institute Helmholtz Centre for Polar and Marine Research (AWI)

In this work, the AWI data source refers to the group of observations that were provided to OC-CCI project by Astrid Bracher. These are bio-optical observations collected during several cruises in the Atlantic and Pacific Oceans. All data were available through the PANGAEA repository. Observations of concentration of chlorophyll-a, and 1nm spectrally resolved remote sensing reflectances and algal pigment absorption coefficient were considered. The methods for these observations are described by Taylor et al. (2011). For chlorophyll, data from the following cruises were used: ANT-XXIV/1, ANT-XXIV/4, ANT-XXVI/4 and MSM18/3 (doi.pangaea.de/10.1594/PANGAEA.847820), SO202/2 (doi.pangaea.de/10.1594/PANGAEA.820607), ANT-XXVII/2 (doi.pangaea.de/10.1594/PANGAEA.848590), ANT-XXV/1 (doi.pangaea.de/10.1594/PANGAEA.819099), ANT-XXVIII/3 and SO218 (doi.pangaea.de/10.1594/PANGAEA.848591). Concerning remote sensing reflectances, the observations taken during cruises ANT-XXIV/4 and ANT-XXVI/4 (doi.pangaea.de/10.1594/PANGAEA.847820), and cruise ANT-XXV/1 (doi.pangaea.de/10.1594/PANGAEA.819099) were gathered. The remote sensing reflectances were corrected for the bidirectional nature of the light field (Morel and Gentili, 1996; Morel et al., 2002). The absorption coefficients were taken during cruise SO202/2 (doi.pangaea.de/10.1594/PANGAEA.820607), cruise ANT_XXV/1 (doi.pangaea.de/10.1594/PANGAEA.819099) and cruises ANT-XXVI/3 and ANT-XXVIII/3 (doi.pangaea.de/10.1594/PANGAEA.819617). The compiled variables were "chla_hplc", "rrs" and "aph".

### 2.2.13 Bermuda Atlantic Time-series Study (BATS)

BATS is a long-term study by the Bermuda Institute of Ocean Sciences based on regular cruises in the western Atlantic Ocean (Sargasso Sea) since 1988. The cruises at BATS site (~ 31º 40'N, 64º 10'W) sample ocean temperature and salinity, but are focused on biogeochemical variables such as nutrients, dissolved inorganic carbon, oxygen, HPLC of pigments, primary production and sediment trap flux. In this work all the phytoplankton pigment data available from the BATS website (http://bats.bios.edu/bats-data/) were considered, which also included regional and transect cruises not specific to the nominal BATS site. The compiled variables were "chla_hplc" and "chla_fluor".



### 2.2.14 Data provided by Knut Yngve Børsheim (BARENTSSEA)

The BARENTSSEA data source refers to a group of observations that were provided to OC-CCI project by Knut Yngve Børsheim. This collection was developed using data from the archives of the Institute of Marine Research (Norway). It comprises observations of temperature, salinity and chlorophyll-a routinely collected by cruises, mainly in the North Sea, the Norwegian Sea and the Barents Sea between 1997 and 2013. The chlorophyll-a concentration was measured by filtering and extraction using Turner fluorometers. The compiled variable was "chla_fluor".

### 2.2.15 The Fisheries and Oceans Canada database for biological and chemical data (BIOCHEM)

BioChem is an archive of marine biological and chemical data maintained by Fisheries and Oceans Canada (DFO, 2018; Devine et al., 2014). The available observations are from department research initiatives and collected in areas of Canadian interest. Available parameters include pH, nutrients, chlorophyll, dissolved oxygen and other plankton data (species and biomass). Chlorophyll measurements from in vitro fluorometric methods were extracted (from http://www.dfo-mpo.gc.ca/science/data-donnees/biochem/index-eng.html) with close guidance by the BioChem helpdesk, confirming quality and methods. The used data span from 1997 to 2014 and were mainly from the Gulf of St. Lawrence (western North Atlantic). The compiled variable was "chla_fluor".

### 2.2.16 British Oceanographic Data Centre (BODC)

(BODC is the designated marine science data centre for the United Kingdom. The data used in this work derive from a specific request for discrete observations of chlorophyll-a concentration since 1997. Initially, this request was used to compile AMT data (see section 2.2.9). The remaining data comprising observations of chlorophyll-a concentration from fluorometric and HPLC methods, mostly sampled in the North Atlantic, were analysed and added (the dataset string for this data source is "bodc"). Data not flagged with highest quality or without method of measurement were discarded. The compiled variables were "chla_hplc" and "chla_fluor".

### 2.2.17 California Cooperative Oceanic Fisheries Investigations (CALCOFI)

CalCOFI is a partnership of the California Department of Fish & Wildlife, National Oceanic & Atmospheric Administration Fisheries Service and Scripps Institution of Oceanography. CalCOFI has conducted quarterly cruises off southern and central California since 1949. Data collected in the upper 500 meters include: temperature, salinity, oxygen, nutrients, chlorophyll, primary productivity, plankton biodiversity, and biomass. For this work, only observations of chlorophyll-a concentration derived from fluorometric methods flagged with highest quality were used. Data were acquired from the file "CalCOFI_Database_194903-201701_csv_20Sept2017.zip" available at http://www.calcofi.org/new.data/index.php/reporteddata#database. The compiled variable was "chla_fluor".

### 2.2.18 California Current Ecosystem Long-Term Ecological Research (CCELTER)



CCELTER investigates the California Current coastal pelagic ecosystem, with a focus on long term forcing. The CCELTER data includes primary and derived measurements from both Process and CalCOFI-augmented cruises, as well other time series. CCELTER data include variables from the physical environment, biogeochemistry and biological populations/communities. For this work chlorophyll observations measured from discrete bottle samples from CCELTER

Process    cruises    determined    by    extraction    and    bench    fluorometry (http://dx.doi.org/10.6073/pasta/7feb632dabb30f0e79683017721a83c7) were compiled. The compiled variable was "chla_fluor".

### 2.2.19 Center for Integrated Marine Technologies (CIMT)

CIMT was a non-operational program where marine scientists from different disciplines and institutions combine their
efforts on observations directed towards understanding the central California upwelling system. The CIMT archived data includes coastal ocean observations from satellites, shipboard data, moorings and large marine animal movements. For this work, pigment data from discrete bottle samples taken during CIMT monthly cruises were used. Data were acquired from the project website (https://cimt.ucsc.edu/data_portal.htm). The compiled variable was "chla_fluor".

### 2.2.20 CoastColour Round Robin (COASTCOLOUR)

COASTCOLOUR datasets were designed to evaluate the performance of ocean colour satellite algorithms in the retrieval of water quality parameters in coastal waters (Nechad et al., 2015). Three types of COASTCOLOUR datasets are available: 1) a match-up dataset where in-situ bio-optical observations are available simultaneously with a cloud-free MERIS product; 2) an in-situ reflectance dataset where an in-situ reflectance is available simultaneously with an in-situ measurement of chlorophyll-a concentration and/or total suspended matter; and 3) a simulated dataset where reflectances were generated by a
radiative transfer model. This work used the match-up dataset, which includes most of the in-situ measurements, and is available at https://doi.pangaea.de/10.1594/PANGAEA.841950. The match-up dataset provides optical, biogeochemical and physical data collections at 17 sites across the globe. From this dataset, observations of reflectance, chlorophyll a, total suspended matter and IOPs were compiled. The remote sensing reflectances were corrected for the bidirectional nature of the light field (Morel and Gentili, 1996; Morel et al., 2002). The compiled variables were "rrs", "chla_hplc", "chla_fluor",
"aph", "adg", "bbp" and "tsm".

### 2.2.21 European Station for Time series in the Ocean, Canary Islands (ESTOC)

ESTOC is an open-ocean monitoring site located in the eastern North Atlantic subtropical gyre. ESTOC was initiated in 1991 with particle flux measurements, and in 1994 began standard observations of the water column, in addition to the deployment of a current meter mooring. The core parameters measured at ESTOC include salinity, temperature, current
speed, nutrients, chlorophyll, inorganic carbon, particulate organic carbon and nitrogen, and sinking particle flux (Neuer et al., 2007). For this work measurements of chlorophyll a concentration from monthly cruises from 1994 to 2011 were used.



These data were provided to CCI following a specific request. The time of day was unavailable and was set to 12:00:00 (UTC). These observations were flagged with "1" in column "flag_time". The compiled variable was "chla_fluor".

### 2.2.22 Integrated Marine Observing System (IMOS)

IMOS is a national collaborative research infrastructure supported by Australian Government. Since 2006, IMOS operates a
wide range of observing equipment throughout the coastal and open ocean around Australia, making all data openly available to the scientific community, and other stakeholders and users. In this work, the IMOS dataset refers only to a data collection entitled IMOS National Reference Station (NRS) - Phytoplankton HPLC Pigment Composition Analysis, which was acquired from the Australian Ocean Data Network Portal (https://portal.aodn.org.au). This dataset comprises phytoplankton pigment composition measured by HPLC collected as part of the IMOS National Mooring Network - National
Reference Station field sampling. Pigment sampling was conducted on a monthly basis with small vessels at nine sites. The IMOS also hosts the Satellite Remote Sensing Bio-optical Database, which comprises phytoplankton pigment composition measured by HPLC collected as part of a suite of bio-optical parameters from samples collected from research voyages in Australian waters, however for this work, the observations from the IMOS Bio-optical Database were acquired as a subset of the SeaBASS dataset. The compiled variable was "chla_hplc".

### 15    2.2.23 MARineEcosytem DATa (MAREDAT)

MAREDAT database is a global assemblage of pigments measured by HPLC (Peloquin et al., 2013) from combination of 136 independent field datasets, solicited from investigators and databases. The database provides high quality measurements of taxonomic pigments including chlorophylls a and b, 19'-butanoyloxyfucoxanthin, 19'-hexanoyloxyfucoxanthin, alloxanthin, divinyl chlorophyll a, fucoxanthin, lutein, peridinin, prasinoxanthin, violaxanthin and zeaxanthin. The database
is available through PANGAEA (http://doi.pangaea.de/10.1594/PANGAEA.793246). For this work only measurements of Total Chlorophyll a flagged with high quality were used. The time of day was unavailable and was set to 12:00:00 (UTC). These observations were flagged with "1" in column "flag_time". The compiled variable was "chla_hplc".

### 2.2.24 Palmer Station Long-Term Ecological Research (PALMER)

PALMER is a monitoring station located in western Antarctic Peninsula. The Palmer station investigates the marine ecology
of the Southern Ocean with focus on the pelagic marine ecosystem, including sea ice habitats, regional oceanography and nesting sites of seabird predators. The PALMER data include measurements of meteorological, oceanographic, sea ice, predators, nutrients and biogeochemistry, pigments, primary production, zooplankton and microbes parameters. This work used the measurements of chlorophyll analysed by HPLC and fluorometry taken at the Palmer Station (http://dx.doi.org/10.6073/pasta/0624c7d161d3b5486d7ba06c2e50ee21                               and
http://dx.doi.org/10.6073/pasta/dea95430a6ad84ecea023ee1ced650d3) and from the annual cruises off the coast of the

Western Antarctica Peninsula (http://dx.doi.org/10.6073/pasta/4d583713667a0f52b9d2937a26d0d82e and http://dx.doi.org/10.6073/pasta/c479b922d42ace1ce37f9a977e214952). The compiled variables were "chla_hplc", "chla_fluor".

### 2.2.25 SeaDataNet archive (SEADATANET)

SeaDataNet is a Pan-European infrastructure for ocean and marine data management. It aims to develop a standardised system for managing large and diverse data sets collected by oceanographic cruises and automatic observation systems. For this work, discrete chlorophyll-a concentration observations with an "access restriction" set to "academic" and "unrestricted" were acquired from the SeaDataNet platform with guidance from helpdesk. Only data from the "Institute of Marine Research - Norwegian Marine Data Centre (NMD), Norway", which comprised most of the acquired data, were used. All chlorophyll observations were from discrete samples measured by fluorometric, spectrophotometric or HPLC methods, but the exact method was not given. Thus, the observations were marked as "chla_fluor", although some were possibly from HPLC measurements, and were flagged with "1" in a column "flag_chla_method". The compiled variables were "chla_fluor".

### 2.2.26 Data provided by Trevor Platt and Shubha Sathyendranath (TPSS)

In this work, the TPSS data source refers to a group of observations that were provided to this compilation by Trevor Platt and Shubha Sathyendranath. This is a collection of bio-optical in-situ data collected during cruises predominantly in the North West Atlantic, but also from the Indian Ocean, South Pacific and Central Atlantic (see Sathyendranath et al. 2009 for additional details). It comprises measurements of phytoplankton pigments and algal pigment absorption coefficients. The time of day was unavailable and was set to 12:00:00 (UTC). These observations were flagged with "1" in column "flag_time". The compiled variables were "chla_hplc", "chla_fluor" and "aph".

### 2.2.27 Bio-optical data from Tara expeditions (TARA)

The Tara expeditions consist of several cruises around the world, some with durations of several years, designed to study and understand the distribution of planktonic organisms in the world ocean. The discrete observations of remote sensing reflectance and chlorophyll-a concentration from HPLC measurements taken during the Tara "Oceans" (2009-2013) and "Mediterranean" (2014) expeditions were considered in this work. These data were provided to ESA OC-CCI project by Emmanuel Boss and were available in the SeaBASS archive. The remote sensing reflectances were corrected for the bidirectional nature of the light field (Morel and Gentili, 1996; Morel et al., 2002). The compiled variables were "chla_hplc" and "rrs".

### 3 Results

In this work several sets of bio-optical in situ data were acquired, homogenised and merged into a single table. The table is



comprises in situ observations between 1997 and 2017, with a global distribution, and includes the following variables: remote-sensing reflectance ("rrs"), chlorophyll-a concentration ("chla"), algal pigment absorption coefficient ("aph"), detrital and coloured dissolved organic matter absorption ("adg"), particle backscattering coefficient ("bbp"), diffuse attenuation coefficient for downward irradiance ("kd") and total suspended matter ("tsm"). All observations in the table were processed

in such a way that they can be compared directly with satellite-derived ocean-colour data. The table consists of 136,250 rows and 1,286 columns. Each row represents a unique station in space and time, separated from the rest by at least 5 minutes and 200 meters. For each observation in a given station, there are three metadata strings: "dataset", "subdataset" and "contributor". The columns of the table take the form described in Table 1. The data contributors are indicated in Table 2. Regarding spectral variables, all original wavelengths were preserved, which requires a large number of unique wavelengths

to be maintained in the database. No band shifting was performed (though some archived data in some data sources may have been merged with nearby wavelengths) and no minimum number of wavelengths per observation was imposed. This allows further manipulation of the table for different purposes. In the following paragraphs, the table is analysed and the final group of observations is described for each contributing data set; however, the numbers reported here do not reflect the original numbers in each data set, since duplicates across contributing data sets were removed (e.g. NOMAD and others

were removed from MERMAID).

Observations of remote-sensing reflectance are available at 611 unique wavelengths (i.e. columns), between 404.7 nm and 1022.1 nm (Fig. 1). In total there are 59,781 observations (i.e. rows) with remote-sensing reflectance in the table. The total number of observations are partitioned per contributing data sets as follows: AERONET-OC (31,574), BOUSSOLE (17,364), MOBY (5,466), NOMAD (3,326), MERMAID (885), SeaBASS (698), AWI (54), COASTCOLOUR (307) and

TARA (107). Data from AERONET-OC, BOUSSOLE and MOBY correspond to continuous time series, and, hence, the higher number of observations. Data distribution at 44X nm and 55X nm is provided in Fig. 2a and b, respectively. Data were first searched at 445 and 555 nm, and then with a search window up to 8 nm, to include also data at 547 nm. Median values at 44X nm range from 0.003 $m^{-1}$ (AERONET-OC) and 0.009 $m^{-1}$ (MOBY), whereas at 55X nm the median values lie between 0.001 $m^{-1}$ (AWI) and 0.007 $m^{-1}$ (COASTCOLOUR). The observations are unevenly distributed between each month

of the year in both hemispheres, with a higher coverage in summer months (Figure 3). There are fewer data in the Southern Hemisphere than in the Northern Hemisphere (Fig. 3). For additional analysis, "rrs" band ratios were plotted against each other (490:555 versus 412:443, Fig. 4). Most points are within the boundaries of the NOMAD dataset, but some scattered points were found. These points were retained in the table to allow further manipulation with different quality control criteria. Complementary analysis of remote-sensing reflectance data is made when other variables are concurrently available

and discussed below (see Fig. 11 and Fig. 16). The geographic distribution of remote-sensing reflectance observations (Fig. 5) shows a higher number of observations in some coastal regions, such as those of North America and Northern Europe. The central regions of the ocean show a lower number of observations, with the Atlantic Ocean having the highest density in relation to the other oceans. Best geographic coverage is provided by the NOMAD database. Data from SeaBASS are fewer in number but are still important. Data from MERMAID are mainly located along the coasts of Europe, North America, and



the central region of the North Atlantic Ocean. The observations from COASTCOLOUR are concentrated in 17 coastal sites around the world, while AWI data are available for the Atlantic, Pacific and Southern Ocean. TARA data are scattered across several regions, with highest data density in Mediterranean Sea.

For chlorophyll-a concentration, two types of observations were compiled, one measured by fluorometric or spectrophotometric methods ("chla_fluor"), and the other measured by HPLC methods ("chla_hplc"). A comparison of both measurements (Fig. 6), when available at the same station shows good agreement (Trees et al., 1985). As stated before, the analysis was done on the final merged table, thus no data were filtered and the good relation can be explained in part by the quality control implemented by the data providers and curators of repositories such as NOMAD and SeaBASS (Werdell and Bailey, 2005). The total number of rows with concurrent "chla_fluor" and "chla_hplc" is 5344, with contributions from SeaBASS (39 %), TPSS (18%), NOMAD (13 %), PALMER (9%), BATS (6%), COASTCOLOUR (5%), MERMAID (4 %), HOT (4 %), AMT+GeP&CO+BODC+CCELTER+CALCOFI (2 %). The "chla_fluor" observations are available in 61,525 stations (rows), with values ranging from 0.001 to 100 mg m$^{-3}$ (Fig. 7). They are from NOMAD (2,350), SeaBASS (18,122), MERMAID (3,711), ICES (5,421), HOT (702),AMT (164), ARCSSPP (189), BARENTSSEA (7,188), BATS (356), BIOCHEM (4,592), BODC (895), CALCOFI (4,631), COASTCOLOUR (3,322), CCELTER (254), CIMT (204.), ESTOC (100), GEPCO (56), PALMER (2,865), SEADATANET (5,403) and TPSS (1000). The total number of "chla_hplc" observations is 23,550, ranging from 0.002 to 99.8 mg m$^{-3}$ (Fig. 7), with contributions from NOMAD (1,309), SeaBASS (9,478), MERMAID (707), ICES (2,994), HOT (193), GeP&CO (1,536), BOUSSOLE (397), AMT (902), AWI (750), BATS (334), BODC (735), COASTCOLOUR (848), IMOS (103), MAREDAT (1,024), PALMER (1,077), TPSS (1,002) and TARA (161). The combined chlorophyll data set (all chlorophyll data considered, but for a given station, HPLC data were selected if available), has a total of 79,731 observations, with 10 %, 49 %, 41 % respectively from oligotrophic (<0.1 mg m$^{-3}$), mesotrophic (0.1 - 1 mg m$^{-3}$), and eutrophic (>1 mg m$^{-3}$) waters. When compared with the proportions of the world ocean in these trophic classes, 56% oligotrophic, 42% mesotrophic and 2% eutrophic (Antoine et al., 1996), oligotrophic waters are under-represented and eutrophic waters are over-represented in the compilation. The combined chlorophyll data set is unevenly distributed between each month of the year in both Northern and Southern Hemispheres, with higher coverage in summer months (Fig. 3). There are fewer data in the Southern Hemisphere than in the Northern Hemisphere (Fig. 3). The spatial distribution of the chlorophyll values for the combined data set (Fig. 8) shows a good agreement with known biogeographical features, such as lower chlorophyll values in the subtropical gyres, and higher values in temperate, coastal and upwelling regions. Many regions show a good spatial coverage (e.g. Atlantic and Pacific Ocean), while others are less well sampled (e.g. Southern and Indian Oceans). Of the contributing data sets, NOMAD and SeaBASS provide a good spatial coverage in many regions (Fig. 9). Other data sets also provide coverage from several locations across the globe (GEPCO, MAREDAT, TARA). The ICES,MERMAID and BODC data are mainly located along the coastal regions of Europe. The AMT and many AWI data mainly covers the central part of the Atlantic Ocean, other AWI data cover the Atlantic sector and the Amundsen to Bellinghausen Sea of the Southern Ocean and the Western subtropical and tropical Pacific. The SEADATANET, ARCSSPP and BARENTSEA provide coverage for the Arctic region and northern seas of the



North Atlantic. The observations from BIOCHEM and TPSS are mostly concentrated in the eastern coast of North America, while CALCOFI, CCELTER and CIMT provide data for the western coast. The remaining data sets provide observations for fixed locations: PALMER (western Antarctic peninsula), COASTCOLOUR (17 coastal sites across the world), BATS (Bermuda, North Atlantic), BOUSSOLE (Mediterranean), HOT (Hawaii, North Pacific), IMOS (coastal sites around

Australia), ESTOC (Canaries, North Atlantic). Figure 9 shows all data sources that contribute with chlorophyll observations, but many overlap each other, especially around Europe and North America. For additional analysis and as an example of the applications of the compiled data set, the combined chlorophyll data ("chla_fluor" and "chla_hplc") were partitioned into 5º x 5º boxes and for each box the number of observations, average value and standard deviation were computed (Fig. 10 a, b and c, respectively). The number of observations can be very high (>1000) in some boxes along the European and North

American coastlines and relatively low (<20) in oceanic regions. Again there is evidence in the average value map (Fig. 10 b) of well-known biogeographical features, such as the lower chlorophyll in the subtropical gyres and higher values in coastal and upwelling areas. There is a close correspondence between the spatial patterns of the average and standard deviation maps (Fig. 10 b and c), which may be an indicator of the data quality.

Coincident observations of chlorophyll-a concentration and remote-sensing reflectance are available at 3,814 stations. These

observations are mostly from NOMAD (79 %), MERMAID (9 %), COASTCOLOUR (6%), and SeaBASS (5 %). The maximum of three band ratios of remote-sensing reflectance is plotted against chlorophyll-a concentration (Fig. 11). The "chla" values used are the combined HPLC and fluorometric chlorophyll-a and for the "rrs", the closest spectral observation within 2 nm was used. The maximum band ratios were calculated using the maximum value between [rrs(443)/rrs(555), rrs(490)/rrs(555), rrs(510)/rrs(555)] or [rrs(443)/rrs(560), rrs(490)/rrs(560), rrs(510)/rrs(560)] if rrs(555) was not available.

The relationship between maximum band ratio and chlorophyll is close to the NASA OC4 and OC4E v6 standard algorithm (http://oceancolor.gsfc.nasa.gov/cms/atbd/chlor_a) equally based on maximum band ratios, providing confidence in the quality of the compiled data.

The inherent optical properties ("aph", "adg" and "bbp") are available at 550 unique wavelengths between 300 and 850 nm. There is a total of 3,293, 1,654 and 792 observations, for "aph", "adg" and "bbp", respectively. For "aph" the total number of

observations is distributed among NOMAD (1,190), TPSS (966), COASTCOLOUR (593), AWI (458), SeaBASS (14) andMERMAID (72).  For "adg" the contributions are as follows: NOMAD (1,079), COASTCOLOUR (531), SeaBASS (11) and MERMAID (33). The "bbp" observations come from NOMAD (371), COASTCOLOUR (154), SeaBASS (32) and MERMAID (235). Data distribution of "aph", "adg" and "bbp" at 44X nm and 55X nm for each data set is provided in Fig. 12 a - f. Median values of "aph", "adg" and "bbp" at 44X and 55X nm for each data set are summarized in Table 3. For

additional analysis, the following band ratios for the absorption coefficients were calculated: aph(490)/aph(443), aph(412)/aph(443), adg(443)/adg(490) and adg(412)/adg(443). Data within 2 nm of the wavelengths were used to maximize the number of points. The distribution of the ratios is shown in Fig. 13. Several observations were found to be outside the thresholds used in the IOCCG report 5 for quality control (see dotted vertical black lines in Fig. 13). These points are




highlighted here for information, but retained in the database, as these were mostly from NOMAD and there was an interest to preserve this data set as a whole. Also, not discarding these data allows further manipulation with different quality control criteria. On the annual scale, the observations of the inherent optical properties are strongly underrepresented in the Southern Hemisphere where there is a complete absence of data in several months of the year (Fig. 3). Overall, the geographic
coverage for observations of "aph", "adg" and "bbp" (Fig. 14) is poor, with most open ocean regions not being sampled, except for the Atlantic Ocean. Small clusters of data are located in particular coastal regions.

Finally, for the diffuse attenuation coefficient for downward irradiance ("kd") there are 25 unique wavelengths between 405 and 709 nm. There is a total of 2,454 observations, from NOMAD (2,266), SeaBASS (118) and MERMAID (70). Data distribution of "kd" at 44X nm and 55X nm for each data set is shown in Fig. 12 g and h. No "kd" data at these wavelengths
were available for the SeaBASS data set (only at 490 nm). Median values of "kd" at 44X nm span between 0.08 m$^{-1}$ (NOMAD) and 0.1 m$^{-1}$ (MERMAID), whereas at 55X nm the "kd" values are approximately 0.1 m$^{-1}$ (NOMAD and MERMAID). NOMAD provides the best geographical coverage (Fig. 15), with a higher coverage in the Atlantic, compared with other oceans. With the exception of the coastal regions of North America and the Japan Sea, most coastal regions are not sampled. In the Northern Hemisphere, "kd" is distributed roughly evenly across all months of the year, but in the
Southern Hemisphere there are few data points during the austral winter and none at all in September (Fig. 3). For total suspended matter ("tsm") there is a total of 1546 observations divided between COASTCOLOUR (1199) and MERMAID (347). The observations of "tsm" are available in a greater number in the Northern Hemisphere (Fig. 3) and are distributed across several coastal regions around Europe, Mediterranean Sea, China Sea, Indonesia and Australia (Fig 15).

Although most of the stations with concurrent variables are from the NOMAD data set, for completeness, an examination of
bio-optical relationships is provided (Fig. 16). The relation between "aph" at 443 nm and chlorophyll-a (Fig. 16 a) agrees with Bricaud et al. (2004). A total of 2,953 points exists with these two variables available (34 % from NOMAD, 32 % from TPSS, 11 % from AWI, 11% from COASTCOLOUR and remaining 12 % from MERMAID and SeaBASS). The relation between the sum of "aph" and "adg" at 443 nm and "rrs" at 443 nm (Fig. 16 b), shows a similar dispersion, with the exception of some scattered points, to an equivalent analysis on the IOCCG report 5 (see their Fig. 2.3). Again, the scattered
data were retained in the final table to preserve the NOMAD data set. A total of 1112 points exists for which these three variables are available (97 % from NOMAD). The relation between the ratio rrs(490)/rrs(555) and kd(490) (Fig. 16 c) shows a good agreement with the NASA KD2S standard algorithm (http://oceancolor.gsfc.nasa.gov/cms/atbd/kd_490). A total of 2,280 points exist for which these three variables are available (93 % from NOMAD). The relation between the ratio rrs(490)/rrs(555) and "bbp" at 555 nm (Fig. 16 c) shows a good agreement with the relation suggested by Tiwari and
Shanmugam (2013). A total of 365 points exist for which these three variables are available (89 % from NOMAD).



## 4 Conclusions

In this work, a compilation of bio-optical in situ data is presented, resulting from the acquisition, homogenization and integration of several sets of data obtained from different sources. The compiled data have a global coverage and span the period from 1997 to 2018. Minimal changes were made on the original data, other than the ones occurring from conversion to standard format and quality control. In situ measurements of the following variables were compiled: remote-sensing reflectance, chlorophyll-a concentration, algal pigment absorption coefficient, detrital and coloured dissolved organic matter absorption coefficient, particle backscattering coefficient, diffuse attenuation coefficient for downward irradiance and total suspended matter.

The final set of data consists of a substantial number of in situ observations, available in a simple text table, and processed in a way that could be used directly for the evaluation of satellite-derived ocean-colour data. The major advantages of this compilation are that it merges six commonly-used data sources in ocean-colour validation (MOBY, BOUSSOLE, AERONET-OC, SeaBASS, NOMAD, MERMAID), four data sources developed for ocean-colour applications (AWI, COASTCOLOUR, TPSS and TARA) and 17 additional sets of chlorophyll-a concentration data (AMT, ICES, HOT,GeP&CO, ARCSSPP, BARENTSSEA, BATS, BIOCHEM, BODC, CALCOFI, CCELTER, CIMT, ESTOC, IMOS, MAREDAT, PALMER, SEADATANET) into a simple text table free of duplicated observations. This compilation was initially created with the intention of evaluating the quality of the satellite ocean-colour products from the ESA OC-CCI project, but it can also be used for other purposes, including the validation of retrievals from recent space-borne sensors such as Landsat 8 and Sentinel-2,3. It may also be useful in the preparation of future sensors like NASA PACE. The objective of publishing the compilation is to make it easily accessible by the broader community.

**Author contribution**

The first six authors belong to the ESA OC-CCI team and contributed to the design of the compilation. The remaining authors are listed alphabetically and are data contributors (see their respective data set on Table 2) or individuals responsible for the development of a particular data set (e.g. Jeremy Werdell for NOMAD and Kathryn Barker for MERMAID). All data contributors (listed in Table 2) were contacted for authorization of data publishing and offered co-authorship. In the case of the ICES data set the permission for publishing was given by the ICES team. All the authors have critically reviewed the manuscript.





## APPENDIX A: Notation

| | |
|---|---|
| ad | Detrital absorption coefficient (m$^{-1}$) |
| adg | Detrital plus CDOM absorption coefficient (m$^{-1}$) |
| AERONET-OC | AErosol RObotic NETwork-Ocean Color |
| ag | CDOM absorption coefficient (m$^{-1}$) |
| AMT | Atlantic Meridional Transect |
| ap | Particle absorption coefficient (m$^{-1}$) |
| aph | Algal pigment absorption coefficient (m$^{-1}$) |
| ARCSSPP | Arctic System Science Primary Production |
| AWI | Data collection from Astrid Bracher |
| aw | Pure water absorption coefficient (m$^{-1}$) |
| BARENTSSEA | Data collection from Knut Yngve Børsheim |
| BATS | Bermuda Atlantic Time-series Study |
| bb | Total backscattering coefficient (m$^{-1}$) |
| bbp | Particle backscattering coefficient (m$^{-1}$) |
| bbw | Backscattering coefficient of seawater (m$^{-1}$) |
| BIOCHEM | The Fisheries and Oceans Canada database for biological and chemical data |
| BODC | British Oceanographic Data Centre |
| BOUSSOLE | Bouée pour l'acquisition d'une Série Optique à Long Terme |
| CALCOFI | California Cooperative Oceanic Fisheries Investigations |
| CCELTER | California Current Ecosystem Long Term Ecological Research |
| CDOM | Coulored Dissolved Organic Matter |
| chla | Chlorophyll a concentration (mg m$^{-3}$) |
| chla_fluor | Chlorophyll a concentration determined from fluorometric or spectrophotometric methods (mg m$^{-3}$) |
| chla_hplc | Total chlorophyll a concentration determined from HPLC method (mg m$^{-3}$) |
| CIMT | Center for Integrated Marine Technology |
| COASTCOLOUR | Compilation of data in several coastal sites |
| Es | Surface irradiance (or above-water downwelling irradiance) (mW cm$^{-2}$ μm$^{-1}$) |
| ESA | European Space Agency |
| ESTOC | Estación Europea de Series Temporales del Oceano |
| Fo | Top-of-the-atmosphere solar irradiance (mW cm$^{-2}$ μm$^{-1}$) |



| | |
|---|---|
| GeP&CO | Geochemistry, Phytoplankton, and Color of the Ocean |
| HOT | Hawaii Ocean Time-series |
| HPLC | High-Performance Liquid Chromatography |
| ICES | International Council for the Exploration of the Sea |
| IMOS | Integrated Marine Observing System |
| kd | Diffuse attenuation coefficient for downward irradiance ($m^{-1}$) |
| Lw | water-leaving radiance (or above-water upwelling radiance) ($mW\ cm^{-2}\ \mu m^{-1}\ sr^{-1}$) |
| MAREDAT | Compilation of data in several coastal sites |
| MERIS | Medium Resolution Imaging Spectrometer |
| MERMAID | MERIS Match-up In situ Database |
| MOBY | Marine Optical Buoy |
| MODIS | Moderate Resolution Imaging Spectro-radiometer |
| NASA | National Aeronautics and Space Administration |
| nLw | Normalized water-leaving radiance ($mW\ cm^{-2}\ \mu m^{-1}\ sr^{-1}$) |
| nLw_ex | nLw with a correction for bidirectional effects ($mW\ cm^{-2}\ \mu m^{-1}\ sr^{-1}$) |
| NOMAD | NASA bio-Optical Marine Algorithm Data set |
| OC-CCI | Ocean Colour Climate Change Initiative |
| OLCI | Ocean and Land Colour Instrument |
| PALMER | Palmer station Long-Term Ecological Research |
| rrs | Remote-sensing reflectance ($sr^{-1}$) |
| Rw | Irradiance reflectance (dimensionless) |
| SeaBASS | SeaWiFS Bio-optical Archive and Storage System |
| SEADATANET | Archive of in situ marine data |
| SeaWiFS | Sea-viewing Wide Field-of-view Sensor |
| TARA | Data collection from global transects |
| TPSS | Data collection from Trevor Platt and Shubha Sathyendranath |
| VIIRS | Visible Infrared Imaging Radiometer Suite |



**APPENDIX B: Data availability**

The compiled data are available at https://doi.org/10.1594/PANGAEA.898188. The database is composed of three main tables: table "insitudb_chla.csv" with the observations of "chla_fluor" and "chla_hplc"; table "insitudb_rrs.csv" with observations of "rrs"; and table "insitudb_iopskdtsm.csv" with remaining observations ("aph", "adg", "bbp", "kd" and "tsm").

The rows within the three tables relate to each other via an unique key (column "idx"). The three tables can be viewed conceptually as one table with all data. To help with data manipulation, six auxiliary tables derived from the previous three main tables are provided. The table "insitudb_metadata.csv" contains all available metadata and helps, for example, to find rows (i.e. "idx") with multiple variables (e.g. "rrs" and "chla_fluor"). The table "auxiliary_table_contributors.csv" contains the number of observations per data contributor, variable and dataset. The remaining four tables

("insitudb_rrs_satbands2.csv", "insitudb_rrs_satbands6.csv", "insitudb_iopskdtsm_satbands2.csv" and "insitudb_iopskdtsm_satbands6.csv") contain the spectral data of the main tables (i.e. "insitudb_rrs.csv" and "insitudb_iopskdtsm.csv") aggregated within ±2 nm and ±6 nm, respectively, of SeaWiFS, MODIS AQUA, MERIS, VIIRS and OLCI sensor bands. The tables are generated by assigning, in each row of the main tables (i.e. "insitudb_rrs.csv" and "insitudb_iopskdtsm.csv"),, the closest spectral observation within 2 nm (or 6 nm) of a sensor band. The centre-wavelengths

of each band and sensor used in the generation of the files are the following: SeaWiFS bands 1-8 were centred at [412, 443, 490, 510, 555, 670, 765, 865] nm, respectively; MODIS-AQUA bands 1-9 were centred at [412, 443, 488, 531, 547, 667, 678, 748, 869] nm, respectively; MERIS bands 1-13 were centred at [412, 442, 490, 510, 560, 620, 665, 681, 709, 753, 779, 865, 885] nm, respectively; VIIRS bands 1-5 were centred at [410, 443, 486, 551, 671] nm, respectively; OLCI bands 1-7 were centred at [412, 442, 490, 510, 560, 620, 665] nm. An exception to this procedure was made to confirm that the correct

MOBY data are stored in the files (see Sect. 2.2.1. for discussion on how MOBY wavelengths are stored in the main file). Finally, a "readme" file is provided to help the user. Table 1 shows how the compiled data looks like. It is given the example of a query for available chlorophyll data from subdataset "seabass_car81".

| idx | time | lat | lon | chla_fluor | chla_fluor_dataset | chla_fluor_subdataset | chla_fluor_contributor |
|---|---|---|---|---|---|---|---|
| 30266 | 2002-08-06T09:02:00Z | 10.5 | -64.67 | 0.185 | seabass | seabass_car81 | Frank_Muller-Karger |

Table B1: Example of how the compiled data looks like. It is shown the result if the compilation is queried for the

chlorophyll data from subdataset "seabass_car81".



**Acknowledgements**

This paper is a contribution to the ESA OC-CCI project. This work is also a contribution to project PEst-OE/MAR/UI0199/2014. We would like to thank the efforts of the teams responsible for collection of the data in the field and of the teams responsible for processing and storing the data in archives, without which this work would not be possible. We thank Tamoghna Acharyya and Robert Brewin at Plymouth Marine Laboratory for their initial contribution to this work. We thank the NOAA (US) for making available the MOBY data, and Yong Sung Kim for the help in questions about MOBY data. BOUSSOLE is supported and funded by the European Space Agency (ESA), the Centre National d'Etudes Spatiales (CNES), the Centre National de la Recherche Scientifique (CNRS), the Institut National des Sciences de l'Univers (INSU), the Sorbonne Université (SU), and the Institut de la Mer de Villefranche (IMEV). We thank ACRI-ST, ARGANS and ESA for access to the MERMAID Database (http://hermes.acri.fr/mermaid). We thank Annelies Hommersom, Pierre Yves Deschamps, Gavin Tilstone and David Siegel for allowing the use of MERMAID data for which they are Principal Investigators. We thank the British Oceanographic Data Centre (BODC) for access to AMT data and in particular to Polly Hadziabdic and Rob Thomas for their help in questions about the AMT data set. We thank Victoria Hill, Patrick Holligan, Gerald Moore and Emilio Suarez for the use of AMT data for which they are Principal Investigators. We thank Sam Ahmed, Hui Feng, Alex Gilerson and Brent Holben for allowing the use of the AERONET-OC data for which they are Principal Investigators. We thank also the AERONET staff and site support people. The Australian Integrated Marine Observing System (IMOS) and CSIRO are acknowledged for funding the Lucinda AERONET-OC site. We thank Bob Bidigare, Matthew Church, Ricardo Letelier and Jasmine Nahorniak for making the HOT data available, and the National Science Foundation for support of the HOT research (grant OCE 09-26766). We thank Yves Dandonneau for allowing the use of GeP&CO data. We thank ICES database on the marine environment (Copenhagen, Denmark, 2014) for allowing the use of their archived data, and Marilynn Sørensen for the help with questions about the ICES data set. We thank all ICES contributors for their data. We thank Eric Zettler and SEA Education Association. The CARIACO Ocean Time-Series program also provided significant decade-long bio-optical information used in this study. These data were obtained from NOMAD and SeaBASS. We thank NASA, SeaBASS and the Ocean Biology Processing Group (OBPG) for access to SeaBASS and NOMAD data. We thank NASA for project funding for data collection. We thank Chris Proctor from SeaBASS for his valuable and prompt help in a variety of questions. We are deeply thankful to the data contributors of NOMAD and SeaBASS:  Kevin Arrigo, Mike Behrenfeld, Emmanuel Boss, Chris Brown, Mary Luz Canon, Douglas Capone, Ken Carder, Alex Chekalyuk, Jay-Chung Chen, Dennis Clark, Jorge Corredor, Glenn Cota, Yves Dandonneau, Heidi Dierssen, David Eslinger, Piotr Flatau, Alex Gilerson, Joaquim Goes, Gwo-Ching Gong, Adriana Gonzalez-Silvera, Larry Harding, Jon Hare, Chuanmin Hu, Sung-Ho Kang, Gary Kirkpatrick, Oleg Kopelevich, Sam Laney, Pierre Larouche, Zhongping Lee, Ricardo Letelier, Marlon Lewis, Steven Lohrenz, Antonio Mannino, John Marra, Chuck McClain, Christophe Menkes, Mark Miller, Ru Morrison, James Mueller, Ruben Negri, James Nelson, Norman Nelson, Mary Jane Perry, David Phinney, John Porter, Collin Roesler, David Siegel, Mike Sieracki, Jeffrey Smart, Raymond Smith, James Spinhirne, Dariusz Stramski, Rick Stumpf, Ajit Subramaniam, Chuck Trees, Ronald Zaneveld, Eric Zettler and Richard Zimmerman. For the BIOCHEM data we thank the Fisheries and Oceans Canada and the following data contributors: Diane Archambault, Hughes Benoit, Esther Bonneau, Eugene Colbourne, Alain Gagne, Yves Gagnon, Tom Hurlbut, Catherine Johnson, Pierre Joly, Maurice Levasseur, Jean-Francois Lussier, Sonia Michaud, Patrick Ouellet, Jacques Plourde, Stephane Plourde, Luc Savoie, Michael Scarratt, Philippe Schwab, Michel Starr and François Villeneuve. We also thank Laure Devine for the help in processing the BIOCHEM data set. We thank Ralph Goericke for allowing the use of the CalCOFI and CCELTER data. CalCOFI research is supported by contributions from the participating agencies: The California State Department of Fish and Wildlife, NOAA,National Marine Fisheries Service, Southwest Fisheries Science Center, and the University of California, Integrative Oceanography Division at the Scripps Institution of Oceanography, UCSD. The authors would like to thank the Oceanic Platform of the Canary Islands (PLOCAN) and its staff for making freely available the use




of this ESTOC data set. We thank the following MAREDAT data providers: Robert Bidigare, Denise Cummings, Giacomo DiTullio, Chris Gallienne, Ralf Goericke, Patrick Holligan, David Karl, Michael Landry, Michael Lomas, Michael Lucas, Jean-Claude Marty, Walker Smith, Rick Stumpf, Emilio Suarez, Koji Suzuki, Maria Vernet and Simon Wright. We thank Oscar Schofield, Raymond Smith and Maria Vernet for allowing the use of the PALMER data. Data from the Palmer LTER
data repository were supported by Office of Polar Programs, NSF Grants OPP-9011927, OPP-9632763 and OPP-0217282. We thank the SeaDataNet Pan-European infrastructure for ocean and marine data management (http://www.seadatanet.org). We thank Emmanuel Boss for the TARA data. Funding for the collection and processing of the TARA data set was provided by NASA Ocean Biology and Biogeochemistry program under grants NNX11AQ14G, NNX09AU43G, NNX13AE58G and NNX15AC08G to the University of Maine. We would like to honour the memory of Marcel Wernand and Tiffany Moisan,
authors who contributed to the first version.

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



**TABLES & TABLE CAPTIONS**

| Variable/Column | Description and units |
|---|---|
| idx | Unique key identifying each row |
| time | GMT, <YYYY-MM-DD>T<HH:MM:SS>Z |
| lat | Decimal degree, -90:90, South Negative |
| lon | Decimal degree, -180:180, West Negative |
| depth_water | Sampling depth (m) – all assigned to zero |
| chla_hplc | Total chlorophyll a concentration determined from HPLC method (mg m$^{-3}$) |
| chla_fluor | Chlorophyll a concentration determined from fluorometric or spectrophotometric methods (mg m$^{-3}$) |
| rrs_<band> | Remote-sensing reflectance (sr$^{-1}$) |
| aph_<band> | Algal pigment absorption coefficient (m$^{-1}$) |
| adg_<band> | Detrital plus CDOM absorption coefficient (m$^{-1}$) |
| bbp_<band> | Particle backscattering coefficient (m$^{-1}$) |
| kd_<band> | Diffuse attenuation coefficient for downward irradiance (m$^{-1}$) |
| tsm | Total suspended matter (g m$^{-3}$) |
| etopo1 | Water depth from ETOPO1 (m) |
| chla_hplc_dataset | Metadata string for "chla_hplc" |
| chla_hplc_subdataset | Metadata string for "chla_hplc" |
| chla_hplc_contributor | Metadata string for "chla_hplc" |
| chla_fluor_dataset | Metadata string for "chla_fluor" |
| chla_fluor_subdataset | Metadata string for "chla_fluor" |
| chla_fluor_contributor | Metadata string for "chla_fluor" |
| rrs_dataset | Metadata string for "rrs" |
| rrs_subdataset | Metadata string for "rrs" |
| rrs_contributor | Metadata string for "rrs" |
| aph_dataset | Metadata string for "aph" |
| aph_subdataset | Metadata string for "aph" |
| aph_contributor | Metadata string for "aph" |
| adg_dataset | Metadata string for "adg" |
| adg_subdataset | Metadata string for "adg" |
| adg_contributor | Metadata string for "adg" |
| bbp_dataset | Metadata string for "bbp" |
| bbp_subdataset | Metadata string for "bbp" |
| bbp_contributor | Metadata string for "bbp" |
| kd_dataset | Metadata string for "kd" |
| kd_subdataset | Metadata string for "kd" |
| kd_contributor | Metadata string for "kd" |
| tsm_dataset | Metadata string for "tsm" |
| tsm_subdataset | Metadata string for "tsm" |
| tsm_contributor | Metadata string for "tsm" |
| flag_time | "1" if observation without time (set to 12:00:00 UTC) |
| flag_chl_method | "1" if observation as unknown chlorophyll method |

**Table 1: The standard variables, nomenclatures and units in the final table.**



| Data Source | Description | Data contributors |
|---|---|---|
| Marine Optical Buoy (MOBY) | Daily observations of remote-sensing reflectance, measured by a fixed mooring system, located west of the Hawaiian Island of Lanai. Data compiled between 1997-2012. Data were obtained from the MOBY website. Compiled standard variable: "rrs". | Paul DiGiacomo, Kenneth Voss |
| Bouée pour l'acquisition d'une Série Optique à Long Terme (BOUSSOLE) | High frequency (15 min) observations of remote-sensing reflectance, from a fixed mooring system, located in the Western Mediterranean Sea. Measurements of chlorophyll-a concentration are also available at the mooring locations. Remote-sensing reflectance and chlorophyll-a data were compiled between 2003-2012 and 2001-2012, respectively. Data were provided by David Antoine. Compiled standard variables: "rrs", "chla_hplc". | David Antoine, Vicenzo Vellucci |
| AErosol RObotic NETwork-Ocean Color (AERONET-OC) | Daily observations of remote-sensing reflectance, measured by modified sun-photometers. Data compiled between 2002-2012. Sites included: Abu_Al_Bukhoosh (~25° N, ~53° E) , COVE_SEAPRISM (~36° N, ~75° W), Gloria (~44° N, ~29° E), Gustav_Dalen_Tower (~58° N, ~17° E), Helsinki Lighthouse (~59° N, ~24° E), LISCO (~40° N, ~73° W), Lucinda (~18° S, ~146° E), MVCO (~41° N, ~70° W), Palgrunden (~58° N, ~13° E), Venice (~45° N, ~12° E), WaveCIS_Site_CSI_6 (~28° N, ~90° W). Data were obtained from the AERONET-OC website. Compiled standard variable: "rrs". | Robert Arnone [WaveCIS], Sam Ahmed [LISCO], Vittorio Brando [Lucinda], Dick Crout [WaveCIS], Hui Feng [MVCO], Alex Gilerson [LISCO], Rick Gould [WaveCIS], Brent Holben [COVE-SEAPRISM], Susanne Kratzer [Palgruden], Thomas Schroeder [Lucinda], Heidi M. Sosik [MVCO], Giuseppe Zibordi [Abu Al Bukhoosh & Gloria & Gustav Dalen Tower & Helsinki Lighthouse & Venice] |
| SeaWiFS Bio-optical Archive and Storage System (SeaBASS) | Global archive of in situ marine data from multiple contributors. Bio-optical global data between 1997-2012 were extracted from the SeaBASS website. Pigment data were mostly extracted using "Pigment Search" tool, which provides data directly from the archives. Radiometric data were extracted using "Validation" tool, which only provides in situ data with matchups for ocean colour sensors. | Robert Arnone, Kevin Arrigo, William Balch, Ray Barlow, Mike Behrenfeld, Sukru Besiktepe, Emmanuel Boss, Chris Brown, Douglas Capone, Ken Carder, Francisco Chavez, Alex Chekalyuk, Jay-Chung Chen, Dennis Clark, Herve Claustre, Lesley Clementson, Jorge Corredor, Glenn Cota, Yves Dandonneau, Heidi Dierssen, David Eslinger, Piotr |





| | Compiled standard variables: "rrs", "chla_hplc", "chl_fluor", "aph", "adg", "bbp", "kd". | Flatau, Robert Frouin, Carlos Garcia, Alex Gilerson, Joaquim Goes, Gwo-Ching Gong, Adriana Gonzalez-Silvera, Rick Gould, Larry Harding, Jon Hare, Stan B. Hooker, Chuanmin Hu, Milton Kampel, Sung-Ho Kang, Gary Kirkpatrick, Oleg Kopelevich, Sam Laney, Pierre Larouche, Jesus Ledesma, Zhongping Lee, Ricardo Letelier, Marlon Lewis, Steven Lohrenz, Mary Luz Canon, Antonio Mannino, John Marra, Chuck McClain, Christophe Menkes, Mark Miller, Greg Mitchell, Ru Morrison, James Mueller, Frank Muller-Karger, Ruben Negri, James Nelson, Norman Nelson, Mary Jane Perry, David Phinney, John Porter, Collin Roesler, David Siegel, Mike Sieracki, Jeffrey Smart, Raymond Smith, Heidi Sosik, James Spinhirne, Dariusz Stramski, Rick Stumpf, Ajit Subramaniam, Chuck Trees, Michael Twardowski, Kenneth Voss, Marcel Wernand, Ronald Zaneveld, Eric Zettler, Giuseppe Zibordi, Richard Zimmerman |
|---|---|---|
| NASA bio-Optical Marine Algorithm Data set (NOMAD) | High-quality global data set of coincident bio-optical in situ data. The data set was built upon SeaBASS archive. The current version (Version 2.0 ALPHA, 2008) was used, with an additional set of columns of remote-sensing reflectance corrected for the bidirectional nature of the light field, provided by NOMAD creators. Data compiled between 1997-2007. Compiled standard variables: "rrs", "chla_hplc", "chl_fluor", "aph", "adg", "bbp", "kd". | Robert Arnone, Kevin Arrigo, William Balch, Ray Barlow, Mike Behrenfeld, Chris Brown, Douglas Capone, Ken Carder, Francisco Chavez, Dennis Clark, Herve Claustre, Jorge Corredor, Glenn Cota, David Eslinger, Piotr Flatau, Robert Frouin, Rick Gould, Larry Harding, Stan B. Hooker, Oleg Kopelevich, Marlon Lewis, Antonio Mannino, John Marra, Mark Miller, Greg Mitchell, Tiffany Moisan, Ru Morrison, Frank Muller-Karger, James Nelson, Norman Nelson, David Siegel, Raymond Smith, Timothy Smyth, James Spinhirne, Dariusz Stramski, Rick Stumpf, Ajit Subramaniam, Kenneth Voss |
| MERIS Match-up In situ Database (MERMAID) | Global database of in situ bio-optical data matched with concurrent MERIS Level 2 satellite ocean colour products The "Extract matchup" tool to acquire data was used. Data | Simon Belanger, Jean-Francois Berthon, Vanda Brotas, Elisabetta Canuti, Pierre Yves Deschamps, Annelies Hommersom, Mati Kahru, Holger Klein, Hubert Loisel, |



| | was compiled between 2002-2012. Access has been granted through a signed Service Level Agreement. Compiled standard variables: "rrs", "chla_hplc", "chl_fluor", "aph", "adg", "bbp", "kd". | David McKee, Greg Mitchell, Michael Ondrusek, Michel Repecaud, David Siegel, Gavin Tilstone, Giuseppe Zibordi |
|---|---|---|
| Atlantic Meridional Transect (AMT) | Multidisciplinary programme that makes biological, chemical and physical oceanographic measurements during an annual voyage between the United Kingdom and destinations in the South Atlantic. It has compiled observations of chlorophyll-a concentration between 1997 (AMT5) and 2005 (AMT17). Data were provided by the British Oceanographic Data Centre (BODC). Compiled standard variables: "chla_hplc", "chl_fluor". | Ray Barlow, Stuart Gibb, Victoria Hill, Patrick Holligan, Gerald Moore, Leonie O'Dowd, Alex Poulton, Emilio Suarez |
| International Council for the Exploration of the Sea (ICES) | Database of several collections of data related to the marine environment. It has compiled observations of chlorophyll-a concentration in the northern European Seas, between 1997-2012. Data were provided by the ICES database on the marine environment (2014, Copenhagen, Denmark). Compiled standard variables: "chla_hplc", "chl_fluor". | Not Available |
| Hawaii Ocean Time-series (HOT) | Multidisciplinary programme that makes repeated biological, chemical and physical oceanographic observations near Oahu, Hawaii. Measurements of chlorophyll-a concentration between 1997-2012 were extracted from the project website. Compiled standard variables: "chla_hplc", "chl_fluor". | Bob Bidigare, Matthew Church, Ricardo Letelier, Jasmine Nahorniak |
| Geochemistry, Phytoplankton, and Color of the Ocean (GeP&CO) | Program of in situ data collection aboard merchant ship from France to New Caledonia, between 1999 and 2002. Measurements of chlorophyll-a concentration were obtained from the project website. Compiled standard variables: "chla_hplc", "chla_fluor". | Yves Dandonneau |
| ARCSSPP | "Arctic System Science Primary Production" database. Available from NODC FTP site. Compiled standard variable: "chla_fluor". | Patricia Matrai |
| AWI | Several 2007-2012 cruises in Atlantic, Pacific and Southern Ocean from Astrid Bracher's group at AWI. Provided by Astrid Bracher. Available from PANGAEA. Compiled standard variables: "chla_fluor", "rrs". "aph". | Astrid Bracher |




| BARENTSSEA | Data collection from cruises of the Institute of Marine Research (Norway) mainly around the Barents Sea. Provided by Knut Yngve Børsheim. Compiled standard variable: "chla_fluor" | Knut Yngve Børsheim |
|---|---|---|
| BATS | Data collection from the "Bermuda Atlantic Time-series Study". Available from BATS website. Compiled standard variables: "chla_fluor", "chla_hplc" | Not Available |
| BIOCHEM | The Fisheries and Oceans Canada database for biological and chemical data. Mostly data from Gulf of St. Lawrence. Available from BIOCHEM website. Compiled standard variable: "chla_fluor". | Diane Archambault, Hughes Benoit, Esther Bonneau, Eugene Colbourne, Alain Gagne, Yves Gagnon, Tom Hurlbut, Catherine Johnson, Pierre Joly, Maurice Levasseur, Patrick Ouellet, Jacques Plourde, Luc Savoie, Michael Scarratt, Philippe Schwab, Michel Starr, François Villeneuve, |
| BODC | "British Oceanographic Data Centre". Mainly European Seas. Provided by BODC. Compiled standard variables: "chla_fluor", "chla_hplc" | Not Available |
| CALCOFI | Cruise data from the "California Cooperative Oceanic Fisheries Investigations" program. Available from CalCOFI website. Compiled standard variable: "chla_fluor". | Ralf Goericke |
| CCELTER | Cruise data from "California Current Ecosystem Long Term Ecological Research". Available from CCELTER website. Compiled standard variable: "chla_fluor". | Ralf Goericke |
| CIMT | Sampling from the "Center for Integrated Marine Technology" (California). Available from CIMT website. Compiled standard variable: "chla_fluor". | Raphael Kudela |
| COASTCOLOUR | Quality controlled compilation of bio-optical data in several coastal sites. Available from PANGAEA. Compiled standard variables: "chla_fluor", "chla_hplc", "rrs", "aph", "adg", "bbp", "tsm". | Not Available |
| ESTOC | Sampling from the "Estación Europea de Series Temporales del Oceano" Canary Islands. Provided by Andrés Cianca. Compiled standard variable: "chla_fluor". | Octavio Llinas and Andres Cianca |
| IMOS | "Australian National Reference Stations". Available from the Australian Ocean Data Network (AODN). Compiled standard variable: | Lesley Clementson |





| | | |
|---|---|---|
| | "chla_hplc". | |
| MAREDAT | Quality controlled global compilation of chla HPLC. Available from PANGAEA. Compiled standard variable: "chla_hplc". | Ray Barlow, Robert Bidigare, Herve Claustre, Denise Cummings, Giacomo DiTullio, Chris Gallienne, Ralf Goericke, Patrick Holligan, David Karl, Michael Landry, Michael Lomas, Michael Lucas, Jean-Claude Marty, Walker Smith, Denise Smythe-Wright, Rick Stumpf, Emilio Suarez, Koji Suzuki, Maria Vernet, Simon Wright |
| PALMER | "Palmer station Long-Term Ecological Research" (Antarctica). Available from PALMER website. Compiled standard variables: "chla_fluor", "chla_hplc". | Oscar Schofield, Raymond Smith, Maria Vernet. |
| SEADATANET | Global archive of in situ marine data. Available from SEADATANET website. Compiled standard variable: "chla_fluor". | Not Available |
| TPSS | Compilation of bio-optical data predominantly from the North West Atlantic, but also from the Indian Ocean, South Pacific and Central Atlantic. Provided by Trevor Platt and Shubha Sathyendranath. Compiled standard variables: "chla_hplc", "chla_fluor", "aph". | Trevor Platt, Shubha Sathyendranath. |
| TARA | Data collection from the TARA global transects. Provided by Emmanuel Boss. All data available in SeaBASS. Compiled standard variables: "chla_hplc", "rrs". | Emmanuel Boss |

**Table 2: Original sets of data and data contributors in the final table.**



| | Median "aph" | | Median "adg" | | Median "bbp" | |
|---|---|---|---|---|---|---|
| | 44x nm | 55x nm | 44x nm | 55x nm | 44x nm | 55x nm |
| SeaBASS | 0.0549 | 0.0074 | 0.0711 | 0.0222 | 0.0035 | 0.0025 |
| MERMAID | 0.0282 | 0.0052 | 0.1149 | 0.0286 | 0.0080 | 0.0052 |
| NOMAD | 0.0353 | 0.0046 | 0.0515 | 0.0112 | 0.0030 | 0.0022 |
| COASTCOLOUR | 0.0665 | 0.0096 | 0.1259 | 0.0175 | 0.0047 | 0.0037 |
| AWI | 0.0208 | 0.0032 | – | – | – | – |
| TPSS | 0.0454 | 0.0071 | – | – | – | – |

**Table 3. Summary of median values for "aph", "adg" and "bbp" at 44X and 55X nm for each data set (as shown in Fig. 12 a-f).**
5    **Data was first searched at 445 and 555 nm, and then with a search window up to 8 nm, to include data at 547 nm.**



**FIGURES AND FIGURES CAPTIONS**

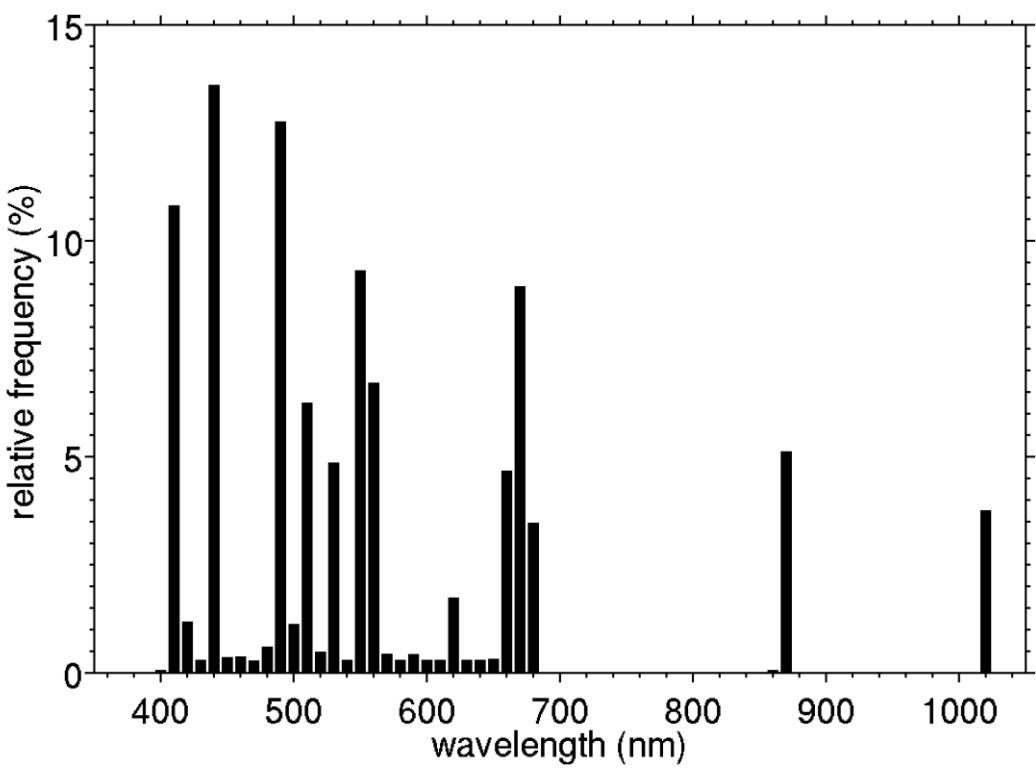

**Figure 1. Relative spectral frequency of remote-sensing reflectance in the final table, using 10 nm wide class intervals, defined as**
5     **the ratio of the number of observations at a particular waveband to the total number of observations at all wavebands, multiplied**
    **by 100 to report results in percentage. Data at a total of 611 unique wavelengths, between 404.7nm and 1022.1 nm, were compiled.**




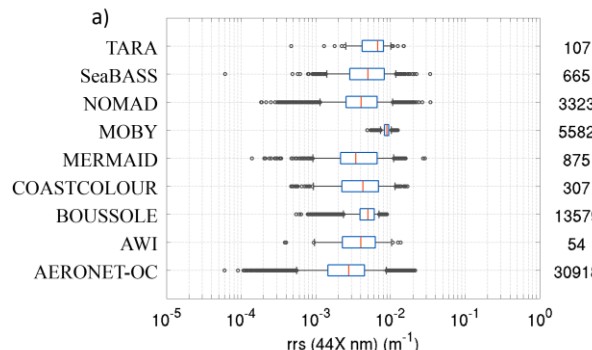
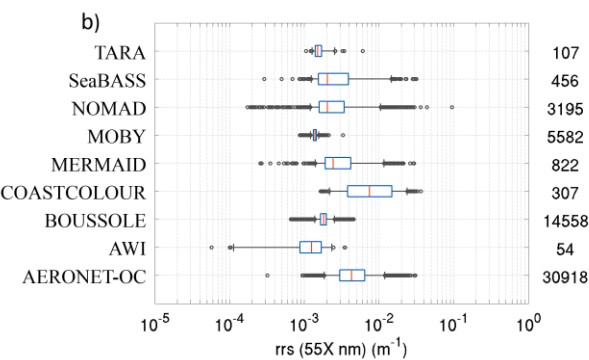

**Figure 2.** The distribution of (a) "rrs" at 44X nm and (b) "rrs" at 55X nm. Data were first searched at 445 and 555 nm, and then with a search window of up to 8 nm, to include data at 547 nm. The black boxes delimit the percentiles 0.25 and 0.75 of the data and the black horizontal lines show the extension of up to percentiles 0.05 and 0.95. The red line represents the median value and the black circles the values below (and above) the percentile 0.05 (0.95). The number of measurements of each data set is reported on the right axis of the graph.

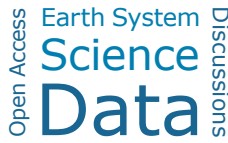

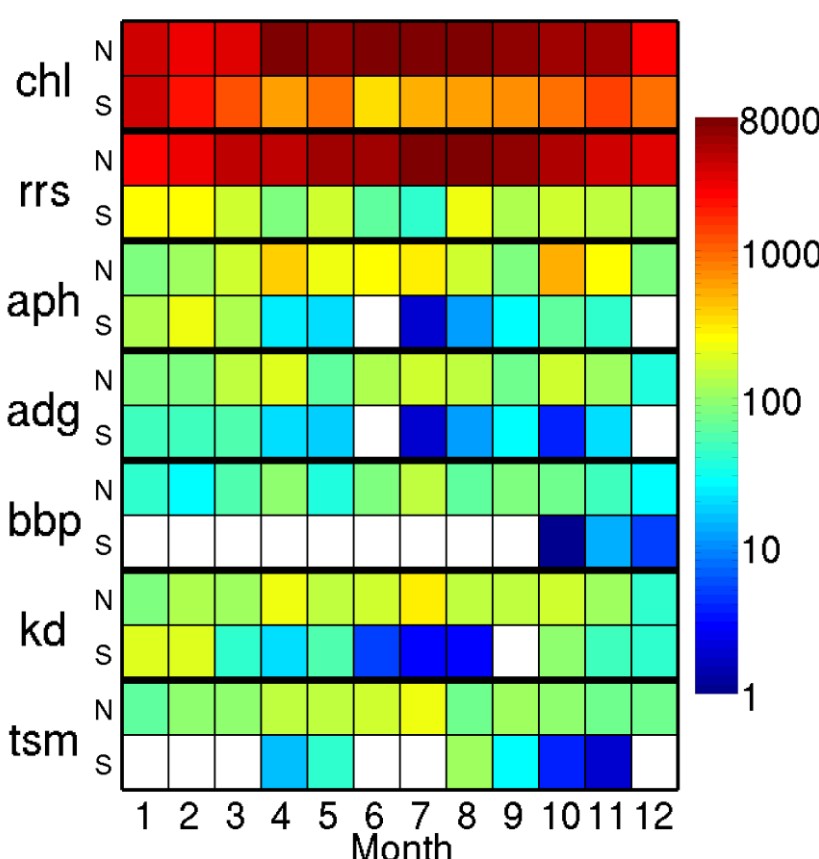

**Figure 3.** Temporal distribution of chlorophyll-a concentration ("chl"), remote-sensing reflectance ("rrs"), algal pigment absorption coefficient ("aph"), detrital plus CDOM absorption coefficient ("adg"), particle backscattering coefficient ("bbp"),the diffuse attenuation coefficient for downward irradiance ("kd") and total suspended matter ("tsm") in the final table. All chlorophyll data were considered, but for a given station, HPLC data were selected if available. Colours indicate the number of stations available for each variable, as a function of month and hemisphere of data acquisition ("N" - Northern Hemisphere; "S" - Southern Hemisphere).




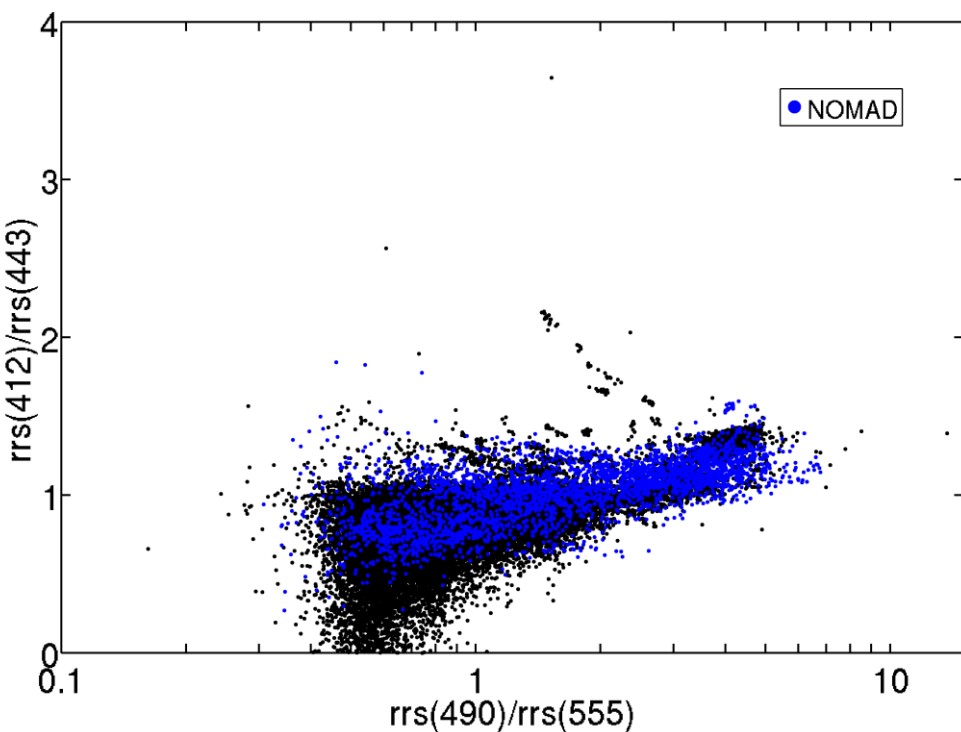

**Figure 4. Ranges of remote-sensing reflectance band ratios (412:443 and 490:555) for all data. The points from the NOMAD data set are shown in blue for reference. To maximize the number of ratios per data set a search window up to 12 nm was used, when the four wavelengths (412, 443, 490, 555) were not simultaneously available. The effect of different search windows was negligible in the ratio distribution.**





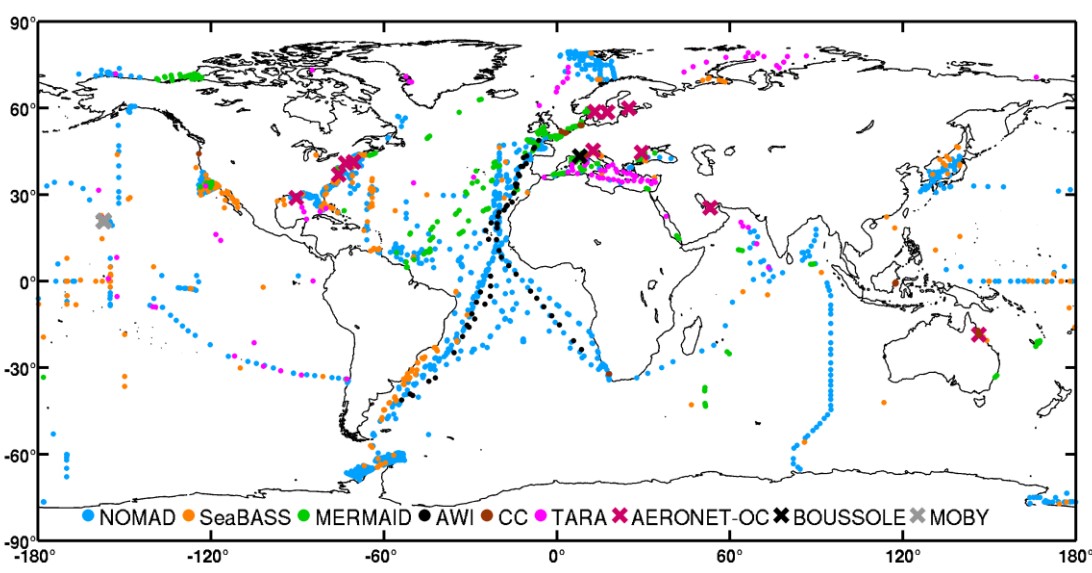

**Figure 5. Global distribution of remote-sensing reflectance per data set in the final table. The data sources are identified with different colours. Points show locations where at least one observation is available. Crosses show sites from where time series data of remote-sensing reflectance are available.**



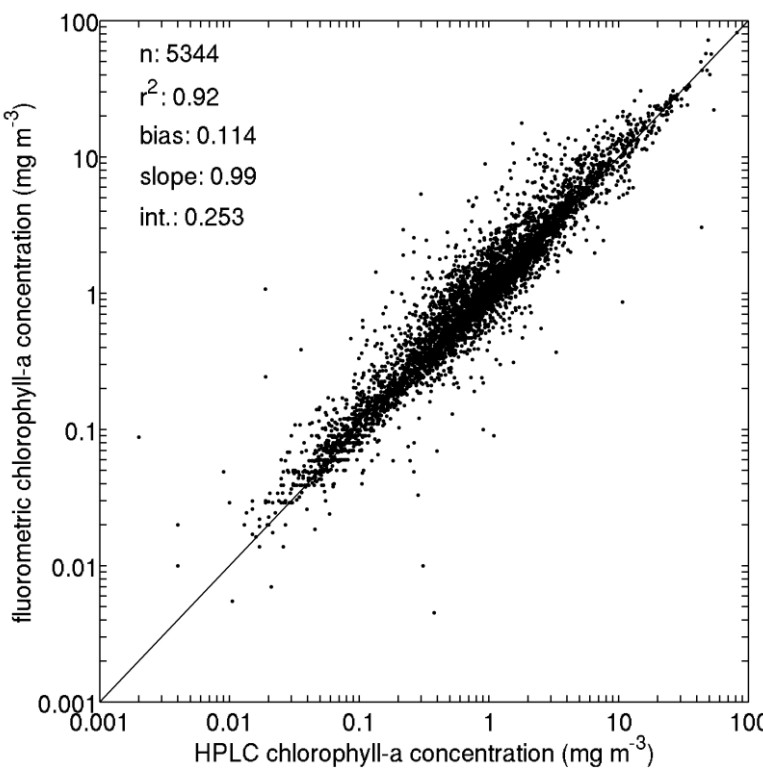

**Figure 6. Comparison of coincident observations of chlorophyll-a concentration derived with different methods ("chla_fluor" and "chla_hplc"). The data were transformed prior to regression analysis to account for their log-normal distribution.**




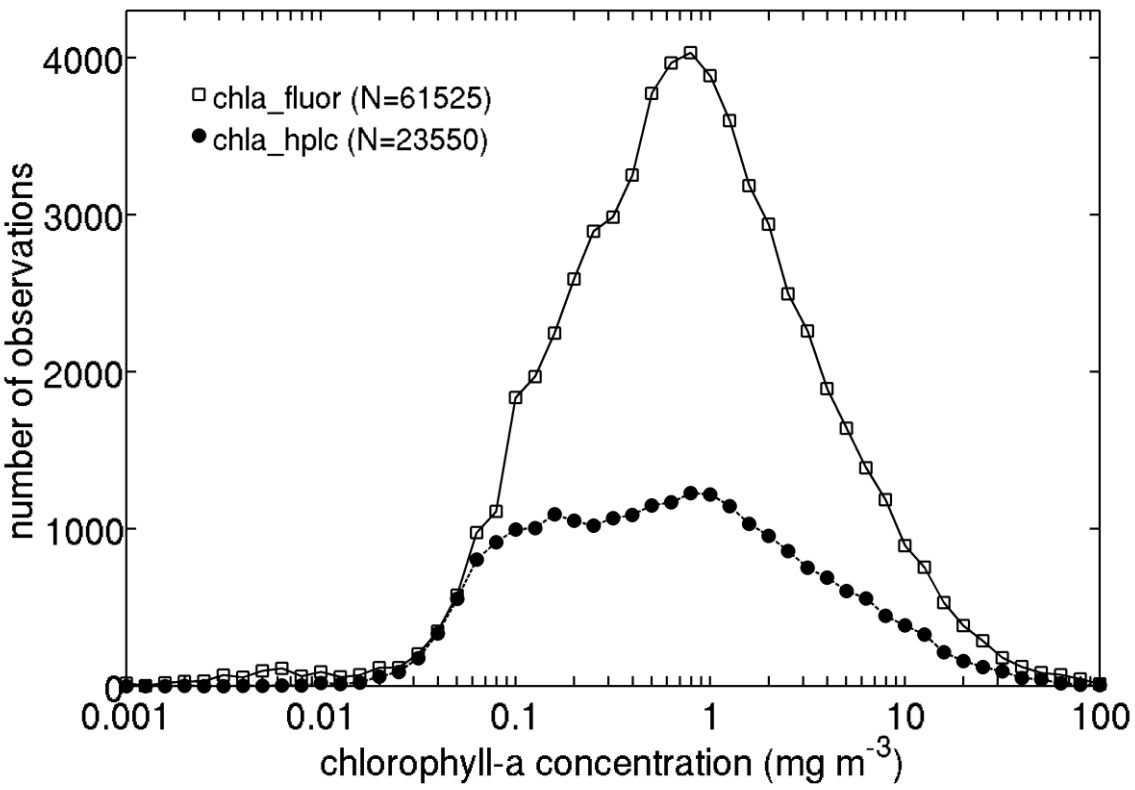

**Figure 7. Number of observations per chlorophyll-a concentration acquired with different methods ("chla_fluor" and "chla_hplc").**





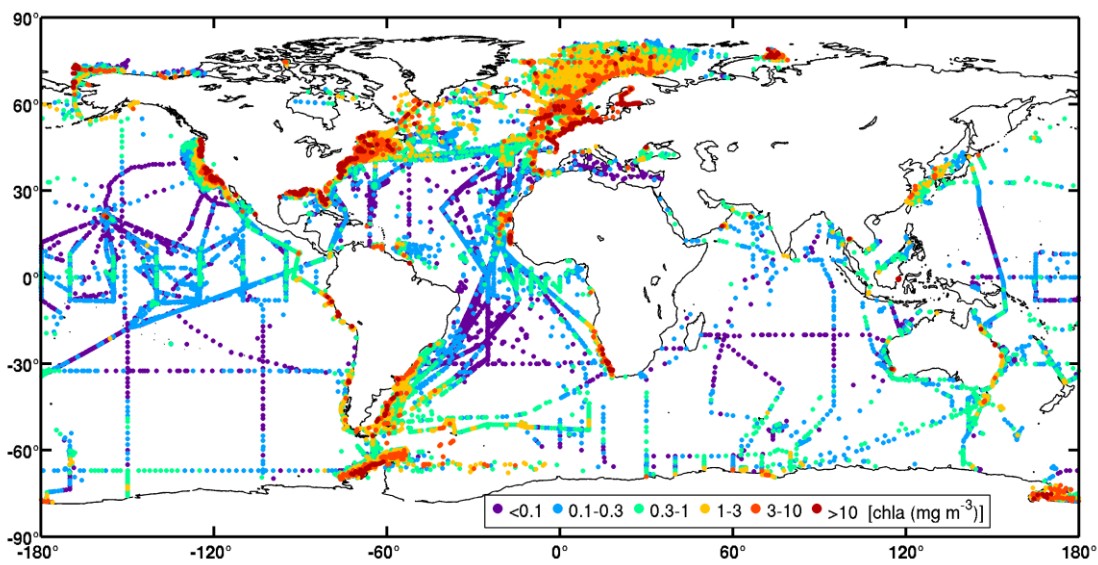

**Figure 8. Global distribution of chlorophyll-a concentration per interval of the observed value. All chlorophyll data were considered, but for a given station, HPLC data were selected if available.**



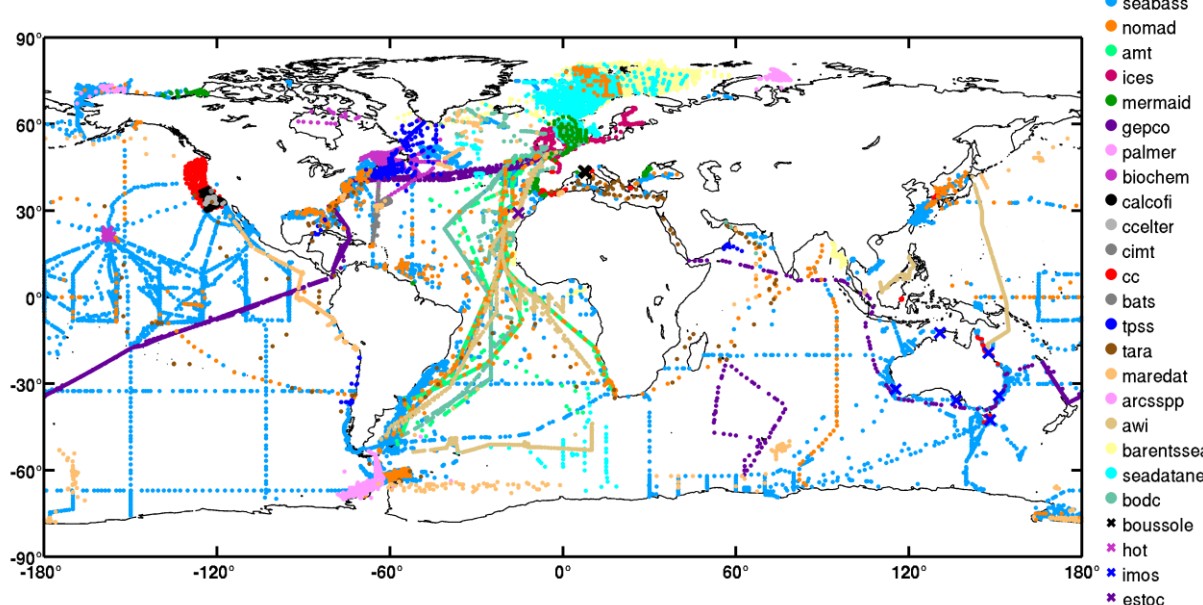

**Figure 9. Global distribution of chlorophyll-a concentration per data set in the final table. All chlorophyll data were considered, but for a given station, HPLC data were selected if available. Crosses show sites from where data of chlorophyll are available in a specific geographic location.**




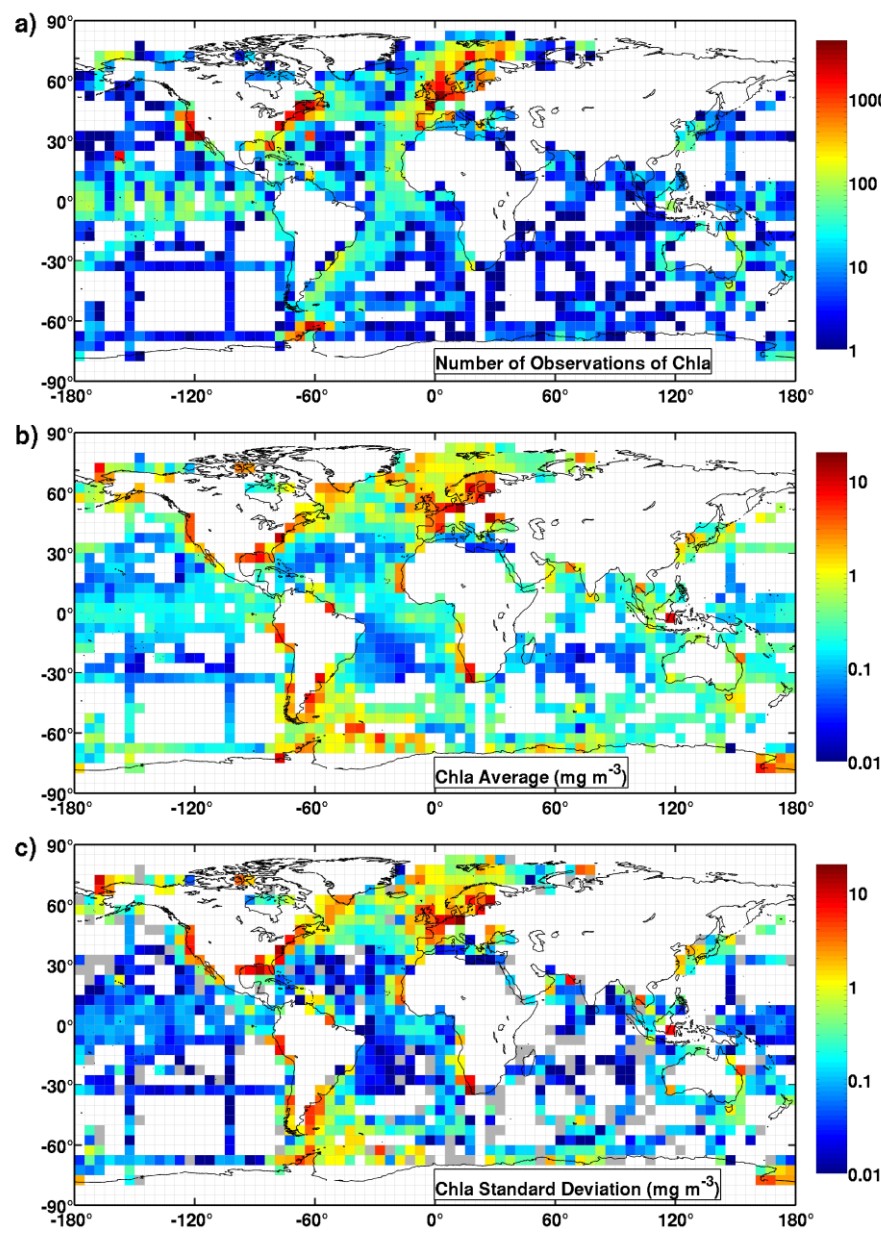

**Figure 10. The chlorophyll-a (mg m$^{-3}$) data partitioned into 5° x 5° boxes showing: (a) number of observations, (b) average value and (c) standard deviation in each box. All chlorophyll data were considered, but for a given station, HPLC data were selected if available. In the standard deviation plot, grey colour boxes represent zero standard deviation (i.e. one observation).**





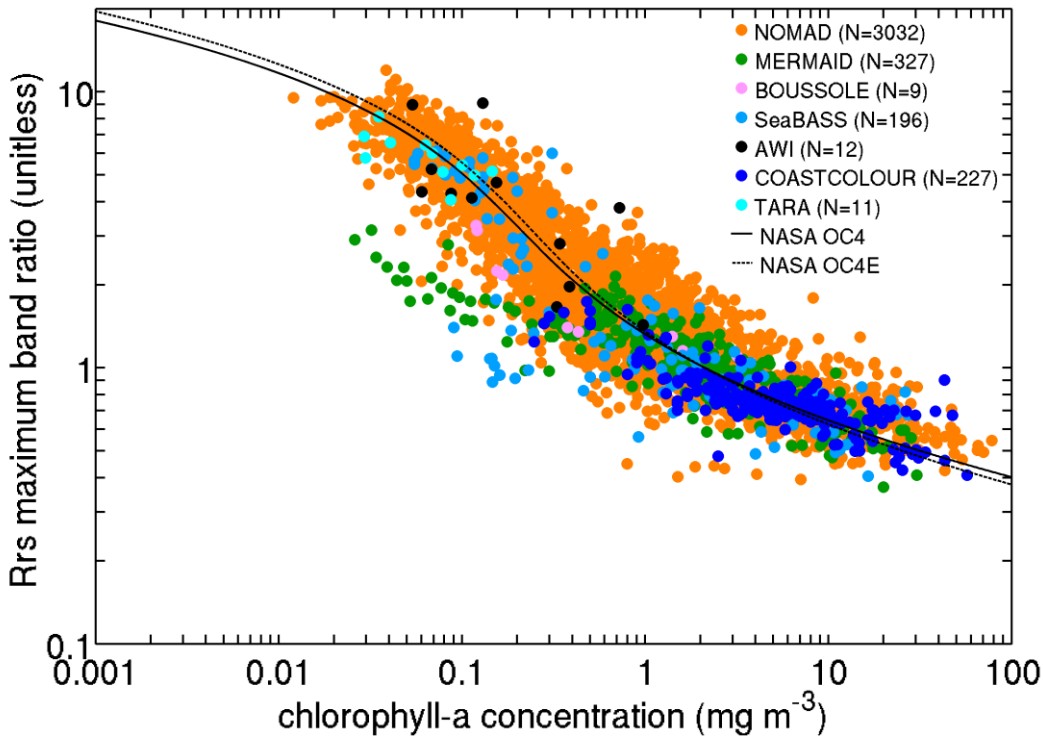

**Figure 11. A remote-sensing reflectance maximum band ratio (as defined in text) ([443,490,510]/555 or [443,490,510]/560 if 555 not available) as a function of chlorophyll-a concentration. All chlorophyll data were considered, but for a given station, HPLC data were selected if available. Data within 2 nm of the wavelengths were used. For reference, the solid and dotted lines show the NASA OC4 and OC4E v6 standard algorithms, respectively (http://oceancolor.gsfc.nasa.gov/cms/atbd/chlor_a). The total number of points was 3,814, of which 79% were from NOMAD.**





**Figure 12. The distribution of: (a) "aph" at 44X nm; (b) "aph" at 55X; (c) "adg" at 44X nm; (d) "adg" at 55X; (e) "bbp" at 44X nm; (f) "bbp" at 55X; (g) "kd" at 44X nm; (h) "kd" at 55Xnm. Data were first searched at 445 and 555 nm, and then with a search window up to 8 nm, to include data at 547 nm. The graphical convention is identical to Fig. 2.**




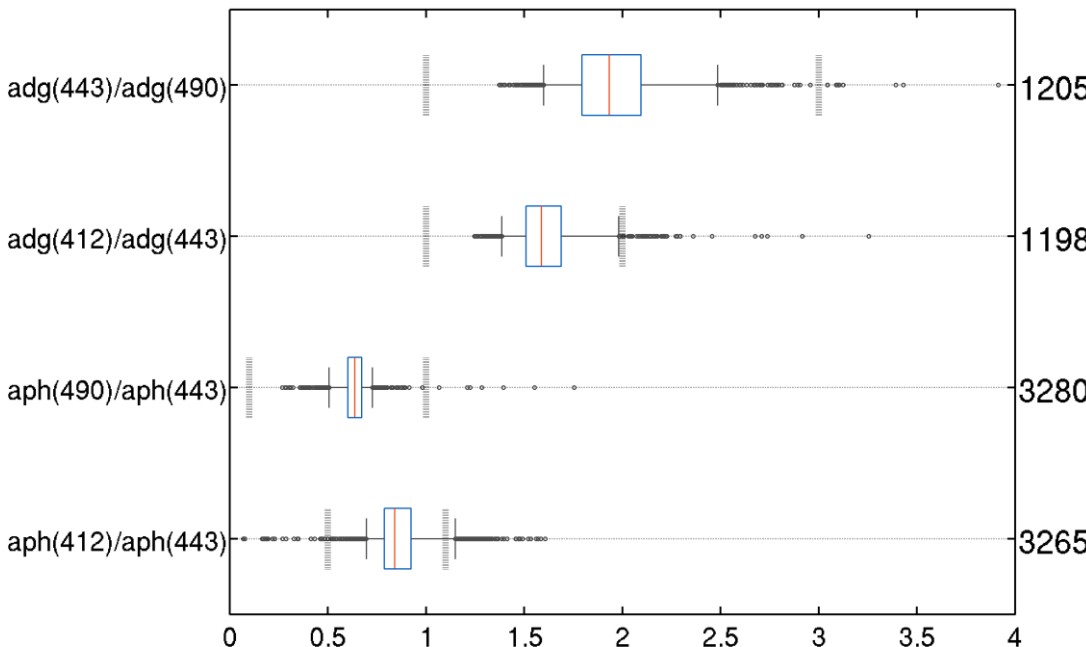

**Figure 13.** The distribution of absorption coefficients band ratios: adg(443)/adg(490), adg(412)/adg(443), aph(490)/aph(443) and aph(412)/aph(443). Data within 2 nm of the wavelengths were used. The graphical convention is identical to Fig. 2. The vertical dashed lines show the lower and upper thresholds used for quality control in the IOCCG report 5. The total number of points for "adg" ratios are divided between NOMAD (89%), COASTCOLOUR (7%), MERMAID (3%) and Seabass (1%). The total number of points for "aph" ratios are divided between NOMAD (36%), TPSS (29%), COASTCOLOUR (18%), AWI (14%), MERMAID (2%) and Seabass (1%).



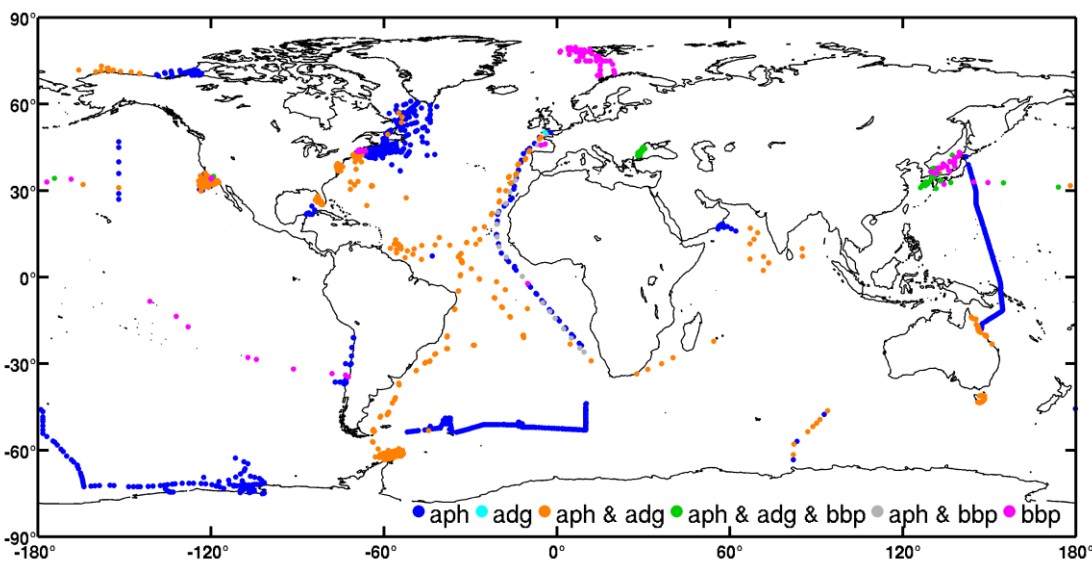

**Figure 14.** Global distribution of observations of inherent optical properties (algal pigment absorption coefficient "aph", detrital plus CDOM absorption coefficient "adg" and particle backscattering coefficient "bbp") in the final table.




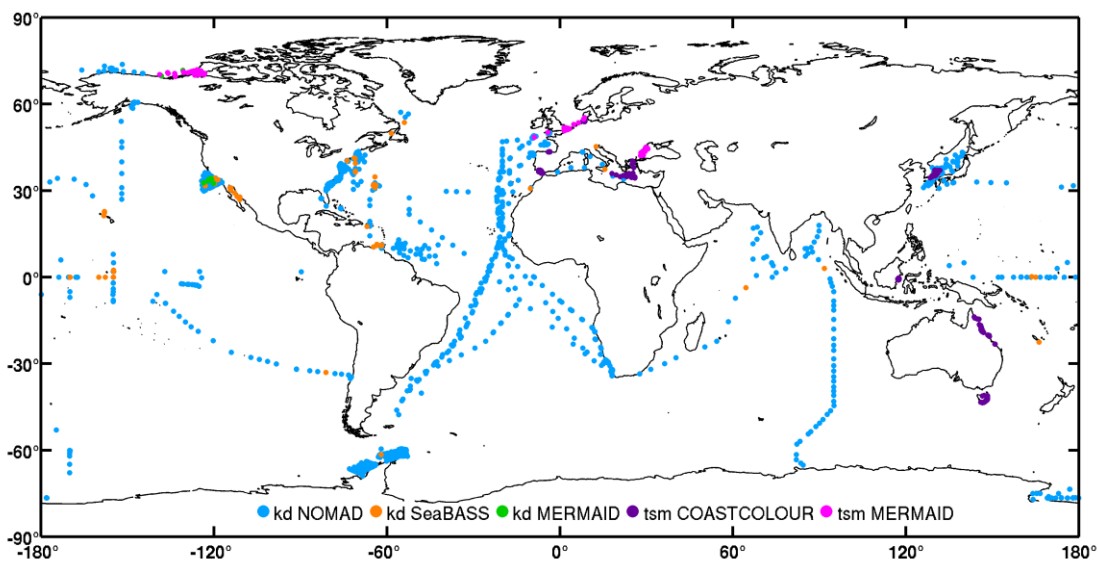

**Figure 15. Global distribution of diffuse attenuation coefficient for downward irradiance ("kd") and total suspended matter ("tsm") per data set in the final table. The "tsm" and "kd" points from MERMAID overlap each other in west Black Sea (~40 ºN 30 ºE) and Arctic (~70 ºN 120 ºW).**

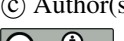



**Figure 16. Examples of bio-optical relationships in the final merged table: (a) aph(443) versus chlorophyll-a. Total number of points (2,953) is divided between AWI (334), COASTCOLOUR (335), MERMAID (214), NOMAD (991),SeaBASS (124) and TPSS (955). For reference the solid line show the regression from Bricaud et al. (2004). (b) [aph(443) + adg(443)] versus rrs(443). Total number of points (1,112) is divided between MERMAID (33) and NOMAD (1,079). (c) [rrs(490)/rrs(555)] versus kd(490). The total number of points (2,280) is divided between MERMAID (62), NOMAD (2,117) and SeaBASS (101). For reference the solid line show the NASA KD2S standard algorithm (http://oceancolor.gsfc.nasa.gov/cms/atbd/kd_490). (d) [rrs(490)/rrs(555)] versus bbp(555). The total number of points (365) is divided between MERMAID (33),NOMAD (324), COASTCOLOUR+SeaBASS (4). For reference the solid line show the relation proposed by Tiwari and Shanmugam (2013). A search window of 2 nm was used for (a) and (b), and a search window of 5 nm was used for (c) and (d) to include data at 560 nm when not available at 555 nm.**