# Peer review of "A compilation of global bio-optical in situ data for ocean-colour satellite applications – version two"

_Earth System Science Data, 2019_

## Referee Comment (RC1) · Anonymous Referee #1 · 17 Mar 2019

This study updates the global high quality ocean-color in-situ datasets, increasing samples of Chl-a, aph, etc., adding new parameter, TSM, and enhance the format to improve operatablity and to keep original information as much as possible. It also includes definition of the variables and explanation of each data source. This revised dataset will help the ocean color community to use the same database for comparison among satellite ocean color algorithms and improve the interoperability among the different satellite datasets. Publication can be strongly recommended but authors should check some small technical issues.

About the new variable, TSM, the tag name,TSM (g/mˆ3) in the document should be consistent to one in the data file (insituab_iopskdtsm.tab), TSS (mg/l). Authors should check again the unit or definition of the TSS in the dataset (insituab_iopskdtsm*.tab).

[Figure]

Some TSM data (especially from "mermaid_BioPotEuroFleets-k0*") seem too low even if the Chl-a range (not so small: 0.28mg/mˆ3 - 17.28mg/mˆ3) and possible variation of the inorganic SM are considered.

The following sentences about the rrs BRDF correction are confusing a little bit:

P5 L31-32: "Thus, for consistency with satellite "rrs" product, only in situ "rrs" that included the latter normalization were included in the compilation."

P12 L22-23: (MERMAID) "Remote-sensing reflectance was calculated by dividing by pi the original irradiance reflectance provided."

I suppose the final value, "rrs", is not the "Remote-sensing reflectance" and it have been calculated by normalizing the "Remote-sensing reflectance". Is that right?

---

## Referee Comment (RC2) · Anonymous Referee #2 · 7 Jun 2019

Very good update of a very useful product. Good data access and helpful data explanation / description.

Small changes suggested:

P4L18: "ARCSSPP,BARENTSSEA" needs a space

P18L29 to P19L1: "The table is comprises in situ observations between 1997 and 2017 . . ." remove the word 'is'?

P41 Figure 3. This reviewer finds Fig 3 very useful but perhaps somewhat confusing. To make their point about predominance of summer-time sampling more strongly, authors might consider regrouping this data into two side-by-side panels, one for NH and a second for SH? At present, the viewer needs to mentally oscillate between NH and

[Figure]

SH as one reads downward by parameter. Also, empty (white) squares indicate no data for that parameter for that month?

---

## Author Response (AR1)

In this PDF it was combined: 1) a point-by-point response to the reviews (as uploaded online), 2) a list of all relevant changes made in the manuscript, 3) and a marked-up manuscript version.

**1 – A point-by-point response to *reviewer #1**

*General comments:* This study updates the global high quality ocean-color in-situ datasets, increasing samples of Chl-a, aph, etc., adding new parameter, TSM, and enhance the format to improve operability and to keep original information as much
5  as possible. It also includes definition of the variables and explanation of each data source.  This revised dataset will help the ocean color community to use the same database for comparison among satellite ocean color algorithms and improve the interoperability among the different satellite datasets. Publication can be strongly recommended but authors should check some small technical issues.

*Comment 1:* About the new variable, TSM, the tag name, TSM (g/m^3) in the document should be consistent to one in the
10 data file (insitudb_iopskdtsm.tab), TSS (mg/l). Authors should check again the unit or definition of the TSS in the dataset (insitudb_iopskdtsm*.tab). Some TSM data (especially from "mermaid_BioPotEuroFleets-k0*") seem too low even if the Chl-a range (not so small: 0.28mg/m^3 - 17.28mg/m^3) and possible variation of the inorganic SM are considered.

*Response:* As for the suspiciously low TSM data from "mermaid_BioPotEuroFleets-k0*", this was an error. Thank you for finding this. Following the reviewer's comment, it was found that all TSM data from "mermaid_BioPotEuroFleets-k0*" was
15 erroneously divided by 1000. The reason is likely that the raw data provided to MERMAID was misspelled "g l-1" instead of "mg l-1". This was now corrected, and the corrected data files were added to PANGAEA (https://doi.pangaea.de/10.1594/PANGAEA.898188). As for the different tag names (TSM in article and TSS in PANGAEA), PANGAEA changed TSM to TSS because that is how TSM is called in their system. It is not possible to change that, so the following comment has been added to the corrected data files in PANGAEA: "tss=tsm in article".

20 *Comment 2:* The following sentences about the rrs BRDF correction are confusing a little bit:

P5 L31-32: "Thus, for consistency with satellite "rrs" product, only in situ "rrs" that included the latter normalization were included in the compilation."

P12 L22-23:  (MERMAID) "Remote-sensing reflectance was calculated by dividing by pi the original irradiance reflectance provided."

25 I suppose the final value, "rrs", is not the "Remote-sensing reflectance" and it have been calculated by normalizing the "Remote-sensing reflectance". Is that right?

*Response:* That is correct, the compiled "rrs" has a BRDF correction. The sentence P5 L31-L32 was changed to: "For consistency with satellite "rrs" product, the latter normalization was applied to the in situ "rrs"."

As for sentence P12 L22-23 in MERMAID section, it was changed to "Remote-sensing reflectance was calculated by
30 dividing by $\pi$ the original "fully-normalized" water-leaving reflectance ("Rw_ex"), which was the water-leaving reflectance (Rw= $\pi$ Lw / Es), with a correction for the bidirectional nature of the light field (Morel and Gentili, 1996; Morel et al., 2002)."

The sentence in P5 L21-23 was deleted. The content of this sentence is now in the new sentence in MERMAID section.

**1 – A point-by-point response to *reviewer #2**

***General comments:*** Very good update of a very useful product. Good data access and helpful data explanation / description. Small changes suggested:

***Comment 1:*** P4 L18: "ARCSSPP, BARENTSSEA" needs a space

5   ***Response:*** space was inserted

***Comment 2:*** P18 L29 to P19 L1: "The table is comprises in situ observations between 1997 and 2017..." remove the word 'is'?

***Response:*** word "is" removed

***Comment 3:*** P41 Figure 3. This reviewer finds Fig 3 very useful but perhaps somewhat confusing. To make their point
10   about predominance of summer-time sampling more strongly, authors might consider regrouping this data into two side-by-side panels, one for NH and a second for SH? At present, the viewer needs to mentally oscillate between NH and SH as one reads downward by parameter. Also, empty (white) squares indicate no data for that parameter for that month?

***Response:*** The main purpose of Figure 3 is to show information about a given variable. Therefore, it seems better that all information of a given variable is provided together (in this case, one in top of the other) not in separate panels. As it is now,
15   the viewer can immediately see for example the difference of "rrs" observations between NH and SH. But if we follow the suggestion and separate panels this hemispheric difference becomes less obvious. For this reason, the suggestion is not followed.

Yes, the empty (white) squares indicate that no data is available. This is now added to the caption of Figure 3.

**2 – A list of all relevant changes made in the manuscript**

The major changes in the manuscript are:

The sentence P5 L31-L32 (in this PDF is P9 L2-L3) was changed.

The sentence P12 L22-23 (in this PDF is P15 L28-L30) was changed.

The sentence in P5 L21-23 (in this PDF is P8 L24-L26) was deleted.

A sentence was added in caption of Figure 3 (in this PDF is P45 L2).

In a few places, the text was improved. These improvements are shown as green-colour "track-changes", whereas the major changes are shown as violet-colour "track-changes".

**3 - Marked-up manuscript version**

[revised manuscript text omitted]